# MODEL MERGING WITH FUNCTIONAL DUAL ANCHORS

## ABSTRACT

Model merging is an efficient post-training strategy for integrating knowledge from multiple finetuned checkpoints of a shared foundation model. Existing methods operate in the parameter space, combining task vectors to mitigate conflicts, but remain constrained by parameter inconsistencies. We propose Functional Dual Anchors (FDAs), a framework that instead models the input-representation space. FDAs are synthetic inputs whose induced gradients align with task vectors, capturing task-specific functional shifts relative to the pretrained model. This perspective bridges joint multi-task training and post-hoc merging, offering both robustness and flexibility. We further introduce a principled initialization scheme and show that FDAs are complementary to parameter-space model merging. Comprehensive experiments demonstrate the effectiveness of FDAs in model merging.

## 1 INTRODUCTION

Model merging has emerged as a promising post-training strategy for integrating knowledge from multiple finetuned checkpoints of foundation models. The core idea is to combine diverse domain knowledge from multiple homologous downstream models into a single unified one (Matena & Raffel, 2022; Jin et al., 2022). Compared to multi-task learning (Ruder, 2017) and continual learning (Wang et al., 2024), model merging is appealing because it consolidates knowledge directly through the parameters of downstream models finetuned from the same pretrained backbone.

However, model merging still faces fundamental challenges due to conflicts arising from diverse task-specific knowledge. Since this knowledge is encoded in the parameters of downstream models, such conflicts inevitably manifest as parameter conflicts. The prevailing paradigm for addressing them is to scale the task

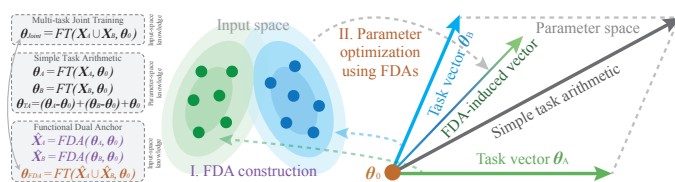

Figure 1: Illustration of our input-space model merging framework using FDAs. On the left, we compare multi-task joint training, task arithmetic and FDA. Inspired by joint training, FDA models the knowledge in the input space. $\boldsymbol{\theta}_A = FT(\boldsymbol{X}_A, \boldsymbol{\theta}_0)$ denotes the model finetuned by the task data $\boldsymbol{X}_A$ from the initial model $\boldsymbol{\theta}_0$ with some loss function.

vectors (Ilharco et al., 2022) (*i.e.*, parameter offsets between these downstream checkpoints and the pretrained model), and then add them back to the pretrained parameters. Within this paradigm, prior works interpret parameter conflicts as task vector conflicts (Yadav et al., 2023; Yu et al., 2024) and propose various adjustment strategies. These methods either exploit intrinsic properties of the parameter space (*e.g.*, magnitude (Yadav et al., 2023; Yu et al., 2024), similarity (Du et al., 2024), orthogonality (Xiong et al., 2024), or subspace structure (Wei et al., 2025b; Gargiulo et al., 2025; Cheng et al., 2025)) or leverage task-specific data to guide adjustments (*e.g.*, entropy measures (Yang et al., 2023; 2025) or representation distributions (Jin et al., 2022; Wei et al., 2025a; Xu et al., 2025)). A unifying characteristic of both approaches is their emphasis on modeling the parameter space, either through structural priors or through data-driven priors.

In contrast to existing approaches, we focus on modeling the input-representation space to mitigate task-specific knowledge conflicts. Rather than directly manipulating parameter offsets, we propose generating synthetic inputs, termed functional dual anchors (FDAs), that can effectively simulate the role of task vectors. An illustration of this idea is provided in Figure 1. Conceptually, this is akin to projecting task-specific knowledge into the input-representation space by constructing inputs that reproduce the downstream model's functional shift relative to the pretrained model. Specifically, for each downstream checkpoint, we construct a set of inputs whose induced gradients on the pretrained parameters align with the corresponding task vector. In this way, FDAs effectively act as the dual of task vectors. While task vectors encode task-specific knowledge in the parameter space,

FDAs capture the analogous knowledge in the input space through their induced gradients. This perspective introduces a new way of thinking about knowledge consolidation. Instead of constraining adjustments to the parameter space, we shift the merging process into the input space, where representations can naturally capture task-specific variations. The key intuition is to bridge the gap between joint multi-task training, where knowledge integration inherently happens in the input space, and model merging, where it is typically confined to the parameter space. By obtaining FDAs for different task vectors, our approach can emulate the effect of joint multi-task training.

To gain an intuitive understanding of FDAs, we compare their optimization trajectories with those of task arithmetic in Figure 2. We treat the obtained FDAs as finetuning data and optimize the model parameters accordingly. As shown in the figure, optimizing with FDAs moves the model closer to the local minima of the loss landscape (computed over eight downstream datasets). While task vectors provide useful guidance from the pretrained model, they quickly drift away from the loss basin, whereas FDAs consistently guide optimization toward more favorable regions. Moreover, by capturing functional shifts in the input space, FDAs offer greater robustness for model merging. Unlike task vectors, which are sensitive to initialization and can drift under different starting points, FDAs exhibit robustness to such variations, facilitating more reliable model merging.



Figure 2: Comparison between task arithmetic and FDAs on the loss landscape of the pretrained across all 8 downstream datasets. FDAs can effectively follow the loss landscape and guide the model toward better local minima.

Another motivation behind FDAs is that modeling the input space is generally easier than modeling the parameter space, as the input space tends to be more structured. The effectiveness of modeling the input space for knowledge transfer is has been extensively explored and empirically validated in the context of dataset distillation (Wang et al., 2018b; Cazenavette et al., 2022), iterative teaching (Liu et al., 2017a; Qiu et al., 2023), dataset condensation (Zhao et al., 2021; Zhao & Bilen, 2023) and continual learning (Shin et al., 2017; Yu et al., 2023). Building on these insights, FDAs provide an alternative perspective on model merging by extending input-space modeling to this setting. Our major contributions are listed as follows:

- Instead of modeling the parameter space, we propose a novel model merging framework that leverages functional dual anchors to model the input–representation space for knowledge encoding.

- Building on theoretical insights from a linear model, we introduce a principled initialization scheme for FDAs, which leads to substantial performance improvements.

- While FDAs can be used independently and yield significant gains, they are also complementary to standard parameter-centric model merging methods, such as TA (Ilharco et al., 2022), TSV (Gargiulo et al., 2025), and WUDI (Cheng et al., 2025). Our empirical results show that incorporating FDAs consistently improves the performance of these parameter-centric approaches.

## 2 A MODEL MERGING FRAMEWORK WITH FUNCTIONAL DUAL ANCHORS

Our model merging framework consists of two stages: (1) FDA construction, and (2) parameter optimization using FDAs. Finally, we discuss the practical implementation of this framework for large-scale foundation models and present the complete procedure in Algorithm 1.

### 2.1 PRELIMINARIES AND BACKGROUND

Before introducing our framework, we briefly recap the formulation of model merging. Consider a foundation model $\varphi$ with pretrained parameters $\boldsymbol{\theta}_0 \in \mathbb{R}^p$ and a collection of downstream finetuned checkpoints with parameters $\{\boldsymbol{\theta}_i\}_{i=1}^m$. The goal of model merging is to derive a merged parameter $\hat{\boldsymbol{\theta}}$ from $\boldsymbol{\theta}_0$ and $\{\boldsymbol{\theta}_i\}_{i=1}^m$ that consolidates knowledge across tasks and achieves multi-task capability without requiring retraining on the original task data. The prevailing approach to model merging is to first compute the task vectors (Ilharco et al., 2022) $\{\boldsymbol{\tau}_i = \boldsymbol{\theta}_i - \boldsymbol{\theta}_0\}_{i=1}^m$, apply adjustments (Yadav et al., 2023; Yu et al., 2024; Wei et al., 2025b) to $\{\boldsymbol{\tau}_i\}_{i=1}^m$, and then add the adjusted task vectors back to the pretrained parameter $\boldsymbol{\theta}_0$. The merged parameter $\hat{\boldsymbol{\theta}}$ is given by $\hat{\boldsymbol{\theta}} = \boldsymbol{\theta}_0 + \sum_{i=1}^m \phi_i(\boldsymbol{\tau}_i)$ where $\phi_i : \mathbb{R}^p \to \mathbb{R}^p$ is introduced to denote possible adjustments of the task vectors $\{\boldsymbol{\tau}\}_{i=1}^m$. In task arithmetic (TA) (Ilharco et al., 2022), $\{\phi_i\}_{i=1}^m$ are linear transformations with a uniform scaling factor between 0 and 1. $\{\phi_i\}_{i=1}^m$ can also take other forms, e.g., the magnitude of parameter values

(Yadav et al., 2023) or the subspace spanned by $\{\boldsymbol{\tau}_i\}_{i=1}^m$ (Xiong et al., 2024). Recently, several works incorporate task-specific entropy measures (Yang et al., 2023; 2025) or representation distribution (Wei et al., 2025a; Xu et al., 2025) to determine $\phi_i$ through iterative optimization. For notational convenience, we use $\varphi(\boldsymbol{\theta}_0)$ to denote the model $\varphi(\boldsymbol{\theta} = \boldsymbol{\theta}_0)$.

Instead of leveraging knowledge in the parameter space, we propose to project the knowledge encoded in checkpoints into the input–representation space. Concretely, we construct a set of synthetic inputs (*i.e.*, FDAs) whose induced gradients on the pretrained model align with task vectors.

## 2.2 FDA CONSTRUCTION: KNOWLEDGE PROJECTION VIA GRADIENT MATCHING

We aim to construct a set of inputs whose induced gradients on the pretrained model align with the task vector. These gradients can be refined by comparing representation discrepancies between the downstream checkpoints $\{\varphi(\boldsymbol{\theta}_i)\}_{i=1}^m$ and the pretrained model $\varphi(\boldsymbol{\theta}_0)$ on the constructed inputs. Formally, assuming the model $\varphi$ operates in a $d$-dimensional input space, we consider a set of $n$ input points $\{\boldsymbol{x}_{ij}\}_{j=1}^n \subset \mathbb{R}^d$ for the downstream checkpoint $\varphi(\boldsymbol{\theta}_i)$. We refer to these points as anchors, as they link $\varphi(\boldsymbol{\theta}_0)$ and $\varphi(\boldsymbol{\theta}_i)$. When these anchors ideally satisfy the following objective, they constitute a set of Functional Dual Anchors (FDAs) for $\varphi(\boldsymbol{\theta}_0)$ and $\varphi(\boldsymbol{\theta}_i)$ (*i.e.*, $\boldsymbol{\tau}_i$):

$$\min_{\boldsymbol{x}_{i1},\ldots,\boldsymbol{x}_{in}} \text{cos\_dist}\left(\nabla_{\boldsymbol{\theta}} \sum_{j=1}^n \text{Dist}\big(\varphi(\boldsymbol{\theta}, \boldsymbol{x}_{ij}), \varphi(\boldsymbol{\theta}_i, \boldsymbol{x}_{ij})\big)\Big|_{\boldsymbol{\theta}=\boldsymbol{\theta}_0}, \boldsymbol{\tau}_i\right), \tag{1}$$

where $\text{cos\_dist}(\boldsymbol{A}, \boldsymbol{B}) = 1 - \frac{\text{vec}(\boldsymbol{A})\text{vec}(\boldsymbol{B})}{\|\boldsymbol{A}\|_F \|\boldsymbol{B}\|_F}$, vec denotes the operation that vectorizes a matrix into a vector in a row-major order, and $\text{Dist}(\cdot)$ denotes a differentiable distance function measuring the representation discrepancy between $\varphi(\boldsymbol{\theta}_0)$ and $\varphi(\boldsymbol{\theta}_i)$. We primarily use cosine distance ($\text{cos\_dist}$), as semantic information is often encoded in direction rather than magnitude (Liu et al., 2017b; 2018). We also evaluate $\ell_1$ and $\ell_2$ distances in Section 5.3. Importantly, the set $\{\boldsymbol{x}_{ij}\}_{j=1}^n$ induces gradients from representation discrepancies that align with the task vector $\boldsymbol{\tau}_i$ in the input-representation space, and thereby serves as the FDAs of $\boldsymbol{\tau}_i$. Correspondingly, we construct a separate set of FDAs $\{\boldsymbol{x}_{ij}\}_{j=1}^n$ for each downstream checkpoint $\varphi(\boldsymbol{\theta}_i)$, *i.e.*, for each task vector $\boldsymbol{\tau}_i$.

**Gradient-based construction for FDAs.** Due to the non-convex nature of Eq. 1, we solve it with gradient descent. We perform gradient-based search in the data space $\mathcal{X}$, where the loss landscape is shaped by fixed model parameters. We refer the process of the gradient-based search in the data space as the *construction process of FDAs*. This process can be formalized as:

$$\boldsymbol{X}_{t+1} = \boldsymbol{X}_t + \eta \cdot \mathcal{U}\left\{\nabla_{\boldsymbol{X}_t} \text{cos\_dist}\Big(\nabla_{\boldsymbol{\theta}} \sum_{j=1}^n \text{Dist}\big(\varphi(\boldsymbol{\theta}, \boldsymbol{x}_{ij}^t), \varphi(\boldsymbol{\theta}_i, \boldsymbol{x}_{ij}^t)\big)\Big|_{\boldsymbol{\theta}=\boldsymbol{\theta}_0}, \boldsymbol{\tau}_i\Big)\right\}, \tag{2}$$

where $\boldsymbol{X}_t = [\boldsymbol{x}_{i1}^t, \ldots, \boldsymbol{x}_{in}^t] \in \mathbb{R}^{d \times n}$ denotes the candidate FDAs at $t$-th iteration; $\mathcal{U}$ denotes the gradient-based optimizer and $\eta$ denotes the update step. While the above gradient-based optimization offers a practical solution in high-dimensional space, it may suffer from slow convergence or limited generalization due to non-convexity. To mitigate these issues, a carefully designed initialization $\boldsymbol{X}_0$ is essential (Glorot & Bengio, 2010; He et al., 2015). We therefore focus on improving initialization to address these optimization challenges. To illustrate how the choice of initialization influences the resulting solution, we begin with an analysis based on a simplified linear model.

**Linear model analysis for initialization.** We consider a linear encoder $\varphi$, *i.e.*, $\boldsymbol{y} = \boldsymbol{W}\boldsymbol{x}$ with $\boldsymbol{W} \in \mathbb{R}^{d \times d}, \boldsymbol{x} \in \mathbb{R}^d$. The pretrained parameters and the downstream parameters on the $i$-th task are denoted by $\boldsymbol{W}_0$ and $\boldsymbol{W}_i$, respectively. Assuming that $\text{Dist}(\boldsymbol{W}_0\boldsymbol{x}, \boldsymbol{W}_i\boldsymbol{x}) = \frac{1}{2}\|\boldsymbol{W}_0\boldsymbol{x} - \boldsymbol{W}_i\boldsymbol{x}\|_2^2$, we analyze the optimization dynamics of a single anchor $\boldsymbol{x}_t$ (with the task index omitted for clarity):

$$\boldsymbol{x}_{t+1} = \boldsymbol{x}_t + \eta\beta_t \Delta\boldsymbol{W}^\top \Delta\boldsymbol{W} \boldsymbol{x}_t, t = 0, \ldots, T-1, \tag{3}$$

where $\Delta\boldsymbol{W} = \boldsymbol{W}_i - \boldsymbol{W}_0$ and $\beta_t = -1/(\|\Delta\boldsymbol{W}\|_F \|\Delta\boldsymbol{W}\boldsymbol{x}_t\|_2 \|\boldsymbol{x}_t\|_2)$. The derivation of Eq. 3 is provided in Appendix A.1. We assume that $\Delta\boldsymbol{W}$ is a full-rank matrix and the eigenvalue magnitudes follow a long-tailed distribution. These assumptions are mild, as empirical evidence shows that parameter updates often follow an approximately low-rank structure (Gur-Ari et al., 2018; Hu et al., 2022; Zhang et al., 2025). Therefore, there exists a spectral decomposition that $\Delta\boldsymbol{W}^\top \Delta\boldsymbol{W} = \boldsymbol{U}\boldsymbol{\Lambda}\boldsymbol{U}^\top, \boldsymbol{U} = [\boldsymbol{u}_1, \ldots, \boldsymbol{u}_d] \in \mathbb{R}^{d \times d}, \boldsymbol{\Lambda} = \text{diag}(\lambda_1, \ldots, \lambda_d)$, with eigenvalues $\lambda_1 > \cdots > \lambda_d > 0$ following a long-tailed distribution. By construction, $\{\boldsymbol{u}_i\}_{i=1}^d$ form a complete basis of the $d$-dimensional space and remain fixed throughout optimization. Thus, we analyze the optimization

trajectory by projecting $x_t$ onto this basis and tracking the dynamics of its coefficients, as formalized in the following proposition. The proof is provided in Appendix A.2.

**Proposition 2.1.** *Under the above setting, for any iteration $t$, $x_t$ can be expressed as the linear combination of $\{u_i\}_{i=1}^d$. Specifically, the coefficient $c_t^i$ associated with basis vector $u_i$ is given by $c_t^i = c_0^i \prod_{j=1}^t (1 - \gamma_j \lambda_i)$, where $\gamma_j = -\eta \beta_j > 0$ and $\beta_j = -1/(\|\Delta W\|_F \|\Delta W x_j\|_2 \|x_j\|_2)$.*

**Remark 2.1.** *For a finite number of iterations $T$, when $|1 - \gamma_j \lambda_i|$ deviates significantly from $1$, then $|c_t^i|$ is dominated by $|1 - \gamma_j \lambda_i|$ due to the exponential growth or decay of the product term, and the effect of initialization is negligible. Conversely, if $|1 - \gamma_j \lambda_i|$ is close enough to $1$ that no exponential growth or decay arises within $T$ iterations, then $|c_t^i|$ remains primarily determined by $|c_0^i|$. This latter case typically occurs when $\lambda_i$ is close to zero.*

**The initialization strategy.** The above analysis suggests that the optimization has almost no effect on components $u_i$ corresponding to near-zero eigenvalues. This motivates an investigation into how initial values of these components affect the convergence of the cosine similarity. Following the above decomposition, we express $\Delta W = Q \Lambda' U^\top$, where $Q$ is an orthogonal matrix and $\Lambda'^2 = \Lambda$. For the $j$-th row, we can write $\Delta W_{j,:} = \sum_{i=1}^d \alpha_{ji} u_i^\top$, $\alpha_{ji} = (Q\Lambda')_{j,i}$. Here, we consider that $Q$ does not amplify the low-energy directions of $\Lambda'$ and $\Lambda$ and assume that the eigenvalues of $\Lambda$ beyond the $k$-th index are near-zero, *i.e.*, $\alpha_{j,i>k} \approx 0$. From Proposition 2.1, that means that $c_t^{i>k} \approx c_0^{i>k}$. We denote the $j$-th row of gradients induced by $x_t$ as $\Delta W_{j,:}^t$. Under the above assumptions, the cosine similarity between $\Delta W_{j,:}$ and $\Delta W_{j,:}^t$ can be approximated as:

$$\frac{\langle \sum_{i=1}^d \alpha_{ji} u_i^\top, \sum_{i=1}^d c_t^i u_i^\top \rangle}{\sqrt{\sum_{i=1}^d \alpha_{ji}^2} \sqrt{\sum_{i=1}^d c_t^{i\,2}}} \approx \frac{\sum_{i=1}^k \alpha_{ji} c_t^i}{\sqrt{\sum_{i=1}^d \alpha_{ji}^2} \sqrt{\sum_{i=1}^k c_t^{i\,2} + \sum_{i=k+1}^d c_0^{i\,2}}} < \frac{\sqrt{\sum_{i=1}^k \alpha_{ji}^2}}{\sqrt{\sum_{i=1}^k \alpha_{ji}^2 + \sum_{i=k+1}^d c_0^{i\,2}}}, \quad (4)$$

where $\Delta W_{j,:}' = \gamma_j x_t^\top = \gamma_j \sum_{i=1}^d c_t^i u_i^\top$, $\gamma_j = -\frac{\partial \mathcal{L}(\varphi)}{\partial (W_0 x_t)_{j,:}}$; $\mathcal{L}$ denotes the finetuning loss. From this expression, the fixed energy in the tail components $\sum_{i=k+1}^d c_0^{i\,2}$ hinders the increase of the cosine similarity at step $t$, which in turn slows down the convergence of the optimization. Moreover, in the idealized case where the first $k$ coefficients are perfectly aligned, the upper bound is given in Eq. 4. Thus, larger initial tail energy leads to lower optimal cosine similarity, whereas smaller tail eigenvalue energy enables faster convergence and results in higher optimal cosine similarity. Given the analysis above, we summarize an initialization principle for FDAs as follows:

**Principle 2.2** (Initialization Principle). *An effective initialization strategy should limit the energy of the initialization point within the tail subspace spanned by the task vector.*

Following the insight from the simplified linear model analysis, we propose two simple yet effective initialization strategies that can control the tail energy.

**Initialization strategy I: Linear Weight Sampling.** We propose to sample the row vectors of the weight matrix as anchor initializations, since they typically also follow a long-tailed spectrum and their total energy is similar to that of the overall $\Delta W$, thereby avoiding excessive tail energy. Specifically, we initialize an anchor $x_{ij} \in \mathbb{R}^d$ by sampling a row of weight matrix $W_i \in \mathbb{R}^{q \times d}$ of $\varphi(\theta_i)$. The process is formalized as $x_{ij} = (W_i)_{l_j,:}, l_j \in \{1, \ldots, q\}$.

**Initialization strategy II: Scaled Gaussian Sampling.** We first draw samples from a standard normal distribution and then scale them using a coefficient $\sigma$. Sampling from a Gaussian ensures that the initialization spans the entire $\mathbb{R}^d$, avoiding zero coefficients in the decomposition along $u_i$. By controlling $\sigma$, we directly constrain the energy of the whole vector, which in turn limits the energy allocated to the tail subspace. The process is formalized as $x_{ij} = \sigma \cdot \tilde{x}_{ij}, \tilde{x}_{ij} \sim \mathcal{N}(0, I_d)$.

## 2.3 PARAMETER OPTIMIZATION: LEVERAGING FDAs FOR MULTI-TASK MODEL MERGING

We leverage the knowledge encoded in FDAs by conducting the dual process of Eq. 1. We first initialize the merged model with the pretrained checkpoint, and then align the output of the model with the downstream checkpoints at all the FDAs. Assume that we have obtained $m$ groups of FDAs $\{x_{ij}\}_{j=1}^n$, one for each $\tau_i$. We then optimize the model parameters with the following objective:

$$\min_{\theta_0} \sum_{i=1}^m \sum_{j=1}^n \text{Dist}\Big(\varphi(\theta_0, x_{ij}), \varphi(\theta_i, x_{ij})\Big), \quad (5)$$

---

**Algorithm 1** Model Merging with Functional Dual Anchors (FDAs)

---

**Input:** Model architecture $\varphi$, pretrained parameters $\boldsymbol{\theta}_0$, downstream parameters $\{\boldsymbol{\theta}_i\}_{i=1}^m$

**Output:** Merged parameter $\hat{\boldsymbol{\theta}}$, FDAs $\{\boldsymbol{x}_{ij}^{(l)}\}_{j=1}^n$, $1 \le i \le m$, $1 \le l \le L$

**for** $l = 1$ **to** $L$ **do**

    `/* --- Stage I: FDA Construction --- */`

    **for** $i = 1$ **to** $m$ **do**

        **Initialization & Optimization:** Initialize $\{\boldsymbol{x}_{ij}^{(l)}\}_{j=1}^n$ using linear weight sampling or scalable Gaussian sampling as starting points and then solve the following objective with gradient descent:

$$\{\boldsymbol{x}_{ij}^{(l)}\}_{j=1}^n = \underset{\boldsymbol{x}_{i1}^{(l)},\ldots,\boldsymbol{x}_{in}^{(l)}}{\arg\min} \; \mathrm{cos\_dist}\Big(\nabla_{\boldsymbol{\theta}^{(l)}} \sum_{j=1}^n \mathrm{Dist}(\varphi^{(l)}(\boldsymbol{\theta}^{(l)}, \boldsymbol{x}_{ij}^{(l)}), \varphi(\boldsymbol{\theta}_i^{(l)}, \boldsymbol{x}_{ij}^{(l)}))\Big|_{\boldsymbol{\theta}^{(l)} = \boldsymbol{\theta}_0^{(l)}}, \boldsymbol{\tau}_i^{(l)}\Big).$$

        Store the optimized anchors $\{\boldsymbol{x}_{ij}^{(l)}\}_{j=1}^n$.

    **end**

    `/* --- Stage II: Parameter Optimization using FDAs --- */`

    Aggregate anchors across tasks $\{\boldsymbol{x}_{ij}^{(l)}\}$.

    Acquire the merged parameter by solving:

$$\hat{\boldsymbol{\theta}}^{(l)} = \underset{\boldsymbol{\theta}^{(l)}}{\arg\min} \sum_{i=1}^m \sum_{j=1}^n \mathrm{Dist}\Big(\varphi^{(l)}(\boldsymbol{\theta}^{(l)}, \boldsymbol{x}_{ij}^{(l)}), \varphi^{(l)}(\boldsymbol{\theta}_i^{(l)}, \boldsymbol{x}_{ij}^{(l)})\Big), \quad \text{from } \boldsymbol{\theta}_0^{(l)}.$$

**end**

**return** $\hat{\boldsymbol{\theta}}$, $\{\boldsymbol{x}_{ij}^{(l)}\}_{j=1}^n$ *for* $1 \le i \le m$, $1 \le l \le L$.

---

which is the *standard adaptation from the pretrained model* (*i.e.*, the first usage of FDAs). The default choice of Dist in Eq. 5 is consistent with that in Eq. 1, and our ablation studies in Sec. 5.3 show that adaptation by FDAs remains robust to different choices of Dist. Please note that in the early optimization stage of Eq. 5, the guidance provided by FDAs approximates to the sum of task vectors. As the optimization proceeds, the guidance provided by FDAs adapts dynamically to the loss landscape of $\boldsymbol{\theta}_t$, while task vectors only prescribe a fixed linear path starting from $\boldsymbol{\theta}_0$.

**Refinement for the merged model**. In particular, we propose the second usage of FDAs by employing them to refine the task vectors obtained from such methods. Given a task vector based merged model $\varphi\left(\boldsymbol{\theta} + \sum_{i=1}^m \phi_i(\boldsymbol{\tau}_i)\right)$, we can refine $\{\phi_i(\boldsymbol{\tau}_i)\}_{i=1}^m$ by minimizing the following objective:

$$\min_{\{\phi(\boldsymbol{\tau}_i)\}_{i=1}^m} \sum_{i=1}^m \sum_{j=1}^n \mathrm{Dist}\Big(\varphi\big(\boldsymbol{\theta} + \sum_{i=1}^m \phi_i(\boldsymbol{\tau}_i), \boldsymbol{x}_{ij}\big), \varphi(\boldsymbol{\theta}_i, \boldsymbol{x}_{ij})\Big). \tag{6}$$

As previously introduced, $\phi_i : \mathbb{R}^p \to \mathbb{R}^p$ denotes possible adjustments of the task vector $\boldsymbol{\tau}_i$. To demonstrate FDAs potential on complementing parameter-centric model merging, we evaluate FDAs on three representative data-free approaches, including TA (Ilharco et al., 2022), TSV (Gargiulo et al., 2025) and WUDI (Cheng et al., 2025). TSV derives $\phi_i(\tau_i)$ by performing Singular Value Decomposition (SVD) and retaining the top components, while WUDI constructs them by reducing the discrepancy between $\sum_{i=1}^m \phi_i(\tau_i)$ and $\{\tau_i\}_{i=1}^m$.

## 2.4 PRACTICAL IMPLEMENTATION

We discuss the practical implementation for Transformer-based foundation models in natural language (Vaswani et al., 2017; Liu et al., 2019), vision (Dosovitskiy et al., 2020; Caron et al., 2021).

**Layer-wise Construction and Adaptation.** The construction process (Eq. 2) involves second-order gradients, which is challenging for the whole foundation models. Instead, we adopt a layer-wise strategy by partitioning the architecture $\varphi$, parameters $\boldsymbol{\theta}_0, \boldsymbol{\theta}_i$ into $L$ parts: $\{\varphi^{(l)}\}_{l=1}^L$, $\{\boldsymbol{\theta}_0^{(l)}\}_{l=1}^L, \{\boldsymbol{\theta}_i^{(l)}\}_{l=1}^L$. For each layer $l$, we construct FDAs for $\boldsymbol{\tau}_i^{(l)} = \boldsymbol{\theta}_i^{(l)} - \boldsymbol{\theta}_0^{(l)}$ and perform adaptation accordingly. Please note that this strategy only requires replacing the entire model in the objectives (Eq. 1, 5, 6) with the corresponding layer-wise components. In our settings, one Resblock is deemed one layer. The overall procedure is summarized as pseudocode in Algorithm 1.

**Shape of FDAs.** Generally, we construct $n$ anchors $\{\boldsymbol{x}_{ij}\}_{j=1}^n \subset \mathbb{R}^d$ for the $i$-th task, where $d$ is the representation dimensionality. For Transformer-based models, the representation space is of the size token_num $\times$ embedding_dim, as they operate at the token level. As embedding_dim is fixed, the shape of FDAs is determined by $n$ and token_num. For vision tasks, we follow the

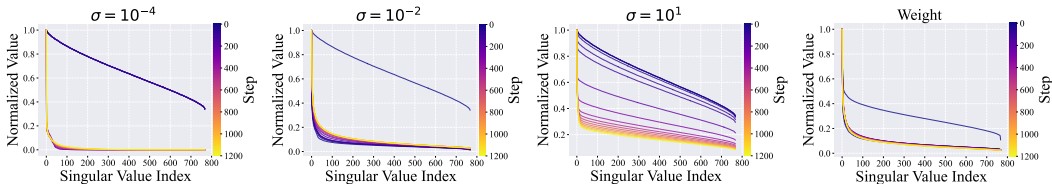

Figure 3: Evolution of Normalized singular values of FDAs in the FDA construction. We visualize the results of FDAs from the 12-th layer of the ViT-B/32 checkpoint on MNIST. $\sigma = 10^1$ denotes FDAs initialized by sampling from $\mathcal{N}(\mathbf{0}, \boldsymbol{I}_d)$ and scaling by $10^1$; "Weight" denotes FDAs initialized from linear weight. FDAs of different initialization schemes tend to evolve into long-tailed structures.

default token_num. For natural language tasks, we set a fixed token_num. Increasing $n$ enlarges the solution space but at the cost of higher computational overhead. We discuss the effect of these hyperparameters in Sec. 5.2 and also list the detailed settings for experiments in Appendix B.3.

**The scale coefficient $\sigma$.** A smaller scaling factor $\sigma$ reduces the tail energy of anchors. However, if $\sigma$ is too small, the head energy is also suppressed, requiring more iterations. A discussion on determining $\sigma$ in practice is in Appendix B.5. For our experiments, we use $\sigma = 0.01$. The effect of $\sigma$ is given in Fig. 4 and Table 4.

# 3 TOWARDS UNDERSTANDING FDA-ENCODED KNOWLEDGE

In this section, we investigate the knowledge encoded by FDAs. We analyze their energy distribution and loss during construction, and compare them with real data in both input–representation and parameter spaces. For analysis, FDAs are constructed from ViT-B/32 (Ilharco et al., 2022) and unfolded into $[n \times \text{token\_num}, \text{embedding\_dim}]$ matrices. Details are in Appendix C.

**Observation 1: FDAs evolve into a long-tailed spectrum structure during optimization.** We perform SVD on the unfolded FDA matrices and normalize singular values by the largest one. From the example in Fig. 3, the normalized tail singular values decays rapidly in construction. This implies that optimization guides FDAs to allocate less energy to the tail, therefore exhibiting a long-tailed structure. The larger tail energy ($\sigma = 10$) results in slower allocation. Furthermore, loss curves (cos_dist) of different initializations in Fig. 4 are consistent with our analysis: the convergence speed first rises and then falls as $\sigma$ decreases

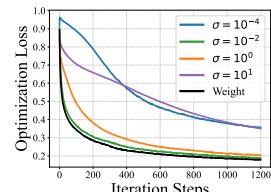

Figure 4: Average loss curves.

from $10^1$ to $10^{-4}$. Notably, initializing FDAs with weights achieves the fastest convergence.

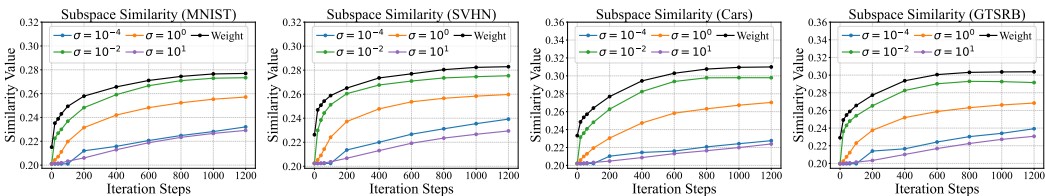

Figure 5: Evolution of subspace similarity of FDAs in the FDA construction. We visualize the results of FDAs from the 12-th layer of the ViT-B/32 checkpoint. $\sigma = 10^1$ denotes the FDAs initialized by sampling from $\mathcal{N}(\mathbf{0}, \boldsymbol{I}_d)$ and scaling by $10^1$; "Weight" denotes the FDAs initialized from linear weight. FDAs of different initialization schemes tend to align the subspace spanned by real data.

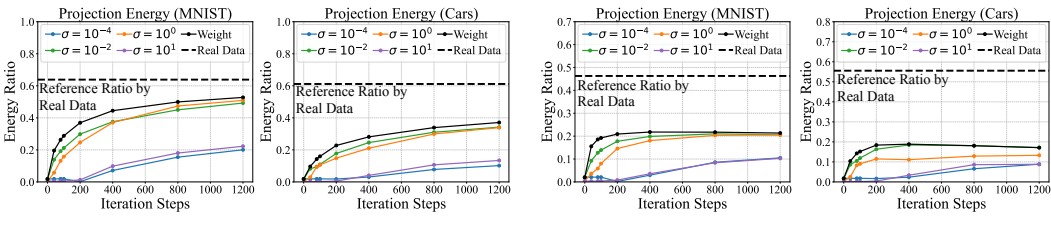

(a) Projection Energy on Pretrained Model.  (b) Projection Energy on Merged Model.

Figure 6: Evolution of projection energy ratio on pretrained model and merged model (TA). We visualize the results of FDAs from the 12-th layer of the ViT-B/32 checkpoints. $\sigma = 10^1$ denotes the FDAs initialized by sampling from $\mathcal{N}(\mathbf{0}, \boldsymbol{I}_d)$ and scaling by $10^1$; "Weight" denotes the FDAs initialized from linear weight. The dashed line indicates the projection energy ratio of task vector induced by real data.

**Observation 2: The high-energy subspace of FDAs gradually aligns with that of real data.** We adopt the features of real task-specific data as reference and also unfolded them. As both real data

(Pope et al., 2021) and FDAs exhibit long-tailed distributions, we measure subspace similarity of top 20% singular vectors via Projection Matrix (Fernando et al., 2013). From Fig. 5, the similarity gradually increases as the optimization proceeds, which is consistent across all datasets. An analysis on whether optimization brings FDAs closer to the manifold of real features is in Appendix C.

**Observation 3: FDAs-induced adaptation increasingly aligns with that induced by real data.** We analyze FDAs by re-projecting them into the parameter space, *i.e.*, the adaptation they induce. We repeat the finetuning procedure in Ilharco et al. (2022) to sample parameter update vectors by real data and project the FDAs-induced adaptation onto their non-negative cone. As shown in Fig. 6, the projection energy onto the sampled cone steadily increases during optimization, both for the pretrained model and merged model by TA. This

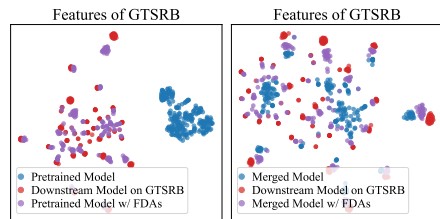

Figure 7: Effects of FDAs on the representations.

indicates that the adaptation induced by FDAs is partially consistent with that of real data. Further, following Yang et al. (2024), we measure the representation discrepancies on real data and observe that FDAs also effectively mitigate representation bias as shown in Fig. 7.

## 4 EXPERIMENTS AND RESULTS

In this section, we first describe the experimental setups and main settings for FDAs. Then we present the performance results. Due to page limits, the remaining setups and results are presented in the Appendix B.

### 4.1 EXPERIMENTAL SETTINGS

**Downstream Models for Merging.** The foundation models in *vision* and *natural language* are both considered. For *vision* tasks, we follow prior works (Ilharco et al., 2022; Yadav et al., 2023) and use eight publicly available domain-specific checkpoints of CLIP Vision Encoder (Radford et al., 2021). All three backbones, ViT-B/32, ViT-B/16, and ViT-L/14, are considered. For *natural language* tasks, following previous works (Yu et al., 2024; Xu et al., 2025), we adopt the downstream checkpoints on GLUE benchmark (Wang et al., 2018a) of RoBERTa-base and RoBERTa-large (Liu et al., 2019). We further extend FDAs to auto-regressive models, WizardMath-13B (Luo et al., 2023) and LLaMA-2-13B-code-Alpaca[1], which are based on LLaMA-2-13B (Touvron et al., 2023), to validate the effectiveness on large models.

**Settings for FDAs.** The layer-wise strategy of FDAs are adopted for the above foundation models. For construction, we use Gaussian and weight initialization. $\sigma$ for Gaussian initialization is fixed at 0.01. For adaptation, two usages (Eq. 5, 6) of FDAs are both considered. For Eq. 6, we consider TA (Ilharco et al., 2022), TSV (Gargiulo et al., 2025), and WUDI (Cheng et al., 2025), where TA serves as a classical baseline, while TSV and WUDI represent recent state-of-the-art approaches. For auto-regressive models, FDAs are constructed and adapted only in each Resblock's feed-forward layer.

**Baseline Methods.** In addition to the mentioned data-free merging methods, we include baselines that use task-specific data to guide adjustments: RegMean (Jin et al., 2022), Fisher merging (Matena & Raffel, 2022), AdaMerging (Yadav et al., 2023), and ProDistill (Xu et al., 2025).

### 4.2 EXPERIMENTS ON VISION AND LANGUAGE MODELS

Table 1, 2 and 3 show the results on the ViT-B/16, RoBERTa-large and auto-aggressive models, respectively. We leave other results in Appendix B. We made the following observations:

**FDAs can effectively leverage existing task-specific knowledge for multi-task model merging.** Specifically, comparing to the dual framework TA, our framework bring a significant improvement: the multi-task performance of pretrained model adapted by FDAs achieves 87.26, compared with 73.94 of TA, representing an improvement of nearly 18%; meanwhile, the average GLUE score achieves 15.4% improvement. Moreover, FDAs also surpass many post-hoc enhancements of vanilla task vectors (Daheim et al., 2023; Yadav et al., 2023; Du et al., 2024; Xiong et al., 2024), while approaching the performance of current state-of-the-art methods.

---

[1] https://huggingface.co/layoric/llama-2-13b-code-alpaca

| Method | SUN397 | Cars | RESISC45 | EuroSAT | SVHN | GTSRB | MNIST | DTD | Avg | Δ |
|---|---|---|---|---|---|---|---|---|---|---|
| Pretrained | 63.80 | 64.60 | 65.70 | 54.50 | 52.00 | 43.30 | 51.70 | 45.10 | 55.00 | - |
| Individual | 78.56 | 87.08 | 96.92 | 99.78 | 97.86 | 99.17 | 99.76 | 82.07 | 92.65 | - |
| RegMean | 70.84 | 75.18 | 83.13 | 94.44 | 90.80 | 82.43 | 98.66 | 60.74 | 82.03 | - |
| Fisher merging | 66.78 | 70.49 | 72.17 | 80.19 | 88.33 | 68.14 | 96.60 | 48.46 | 73.89 | - |
| AdaMerging | 64.30 | 74.37 | 74.63 | 94.89 | 91.19 | 94.94 | 97.95 | 69.63 | 82.74 | - |
| Representation Surgery | **73.60** | 81.50 | 90.40 | 98.50 | 93.20 | 97.40 | 98.90 | **77.00** | 88.80 | - |
| ProDistill | 72.82 | **81.94** | 91.94 | 99.52 | **97.11** | 97.65 | **99.60** | 70.74 | **88.92** | - |
| TA | 62.07 | 66.14 | 74.00 | 76.48 | 88.02 | 73.79 | 98.52 | 52.50 | 73.94 | - |
| TSV | 72.83 | 80.20 | 88.97 | 97.22 | 93.93 | 93.94 | 99.27 | 72.66 | 87.38 | - |
| WUDI | 75.40 | 81.71 | 90.14 | 98.52 | **95.30** | **96.55** | 99.44 | 73.78 | 88.85 | - |
| FDA (Pretrained, Gauss) | 72.54 | 80.62 | 87.75 | 98.44 | 94.31 | 93.43 | 99.38 | 70.11 | 87.07 | +58.30 |
| FDA (Pretrained, Weight) | 73.60 | 80.48 | 88.00 | **99.31** | 94.35 | 93.41 | 99.31 | 70.64 | 87.26 | +58.65 |
| FDA (TA, Gauss) | 73.72 | 81.42 | 88.63 | 98.37 | 94.61 | 94.44 | 99.39 | 71.54 | 87.77 | +13.83 |
| FDA (TA, Weight) | 74.53 | 81.25 | 88.37 | 98.37 | 94.55 | 94.28 | 99.34 | 71.65 | 87.79 | +13.85 |
| FDA (TSV, Gauss) | 74.79 | 82.65 | 89.75 | 98.37 | 94.25 | 94.47 | 99.40 | 73.67 | 88.42 | +1.04 |
| FDA (TSV, Weight) | 74.93 | 81.92 | 89.79 | 98.33 | 94.10 | 93.78 | 99.36 | 73.78 | 88.25 | +0.87 |
| FDA (WUDI, Gauss) | **76.21** | **82.84** | 91.03 | 98.93 | 94.58 | 96.32 | 99.40 | 74.52 | **89.23** | +0.38 |
| FDA (WUDI, Weight) | 76.15 | 82.75 | **91.21** | 98.89 | 94.49 | 96.24 | 99.39 | 74.41 | 89.19 | +0.34 |

Table 1: Performance of merging ViT-B-16 models across eight downstream vision tasks. FDAs (*init model*, *FDAs init*) denotes the choice of the initial model and the initialization strategies for FDAs, respectively. "Δ" denotes the percentage gain compared to the initial model.

| Method | CoLA | SST-2 | MRPC | STS-B | QQP | MNLI | QNLI | RTE | Avg | Δ |
|---|---|---|---|---|---|---|---|---|---|---|
| Pretrained | 0.1679 | 0.4897 | 0.7480 | -0.0471 | 0.3159 | 0.3545 | 0.5054 | 0.4693 | 0.3754 | - |
| Individual | 0.6335 | 0.9001 | 0.9224 | 0.9418 | 0.9055 | 0.8267 | 0.9507 | 0.9222 | 0.8754 | - |
| RegMean | 0.3449 | 0.8922 | 0.5949 | 0.3509 | 0.8045 | 0.5894 | 0.6132 | 0.6534 | 0.6054 | - |
| Fisher merging | 0.2700 | 0.7856 | 0.7517 | 0.2624 | 0.3159 | 0.4385 | 0.5367 | 0.6426 | 0.5004 | - |
| AdaMerging | 0.1027 | 0.9335 | 0.7480 | **0.7432** | 0.3159 | 0.7506 | 0.8578 | 0.6245 | 0.6345 | - |
| ProDistill | **0.4833** | **0.9427** | **0.8655** | 0.7310 | **0.8269** | **0.8122** | **0.8825** | **0.7545** | **0.7873** | - |
| TA | 0.1635 | 0.8716 | 0.7480 | 0.6603 | 0.3159 | 0.6101 | 0.8716 | 0.7366 | 0.5918 | - |
| TSV | 0.4791 | 0.9323 | 0.7459 | 0.6660 | 0.3300 | 0.6750 | 0.7761 | 0.6751 | 0.6599 | - |
| WUDI | 0.4201 | 0.9232 | 0.7487 | 0.7345 | 0.5393 | 0.6430 | 0.5746 | 0.5740 | 0.6447 | - |
| FDAs (Pretrained, Gauss) | 0.3198 | 0.8463 | 0.7790 | 0.6828 | **0.7423** | 0.5605 | 0.6021 | **0.7726** | 0.6632 | +0.2878 |
| FDAs (Pretrained, Weight) | 0.3883 | 0.8911 | **0.7858** | 0.7230 | 0.7410 | 0.5791 | 0.6207 | 0.7329 | 0.6827 | +0.3073 |
| FDAs (TA, Gauss) | 0.4043 | **0.9461** | 0.7692 | 0.7897 | 0.6916 | 0.7190 | 0.7487 | 0.7076 | **0.7220** | +0.3466 |
| FDAs (TA, Weight) | 0.4511 | 0.9404 | 0.7578 | 0.7926 | 0.6518 | **0.7411** | 0.6965 | 0.7148 | 0.7183 | +0.3429 |
| FDAs (TSV, Gauss) | **0.5036** | 0.9438 | 0.7521 | **0.7975** | 0.4128 | 0.7075 | **0.8477** | 0.7365 | 0.7127 | +0.3373 |
| FDAs (TSV, Weight) | 0.5021 | 0.9427 | 0.7490 | 0.7418 | 0.5062 | 0.7292 | 0.8146 | 0.7365 | 0.7153 | +0.3399 |
| FDAs (WUDI, Gauss) | 0.4841 | 0.9404 | 0.7647 | 0.7645 | 0.6778 | 0.7004 | 0.5911 | 0.6643 | 0.6984 | +0.3230 |
| FDAs (WUDI, Weight) | 0.4848 | 0.9392 | 0.7573 | 0.7546 | 0.6979 | 0.7072 | 0.5656 | 0.6643 | 0.6964 | +0.3210 |

Table 2: Performance of merging RoBERTa-large models across eight NLU tasks. FDAs (*init model*, *FDAs init*) denotes the choice of the initial model and the initialization strategies for FDAs, respectively. "Δ" denotes the percentage gain compared to the initial model.

**FDAs shows that projecting task-specific knowledge into the input-representation space surfaces more task-specific knowledge for merging.** Specifically, although FDAs and data-free parameter-centric methods leverage the same task-specific knowledge, FDAs still improve the merged models by these methods. The average improvement via FDAs on TA, TSV, and WUDI is nearly $5.10\%$ on ViT-B/16, and about $13\%$ on RoBERTa-Large. For the auto-regressive model, as we only adapt for feed-forward network, FDA still achieves $10\%$ improvement on TA.

| Method | GSM8K | MATH | MBPP | HEval | Avg |
|---|---|---|---|---|---|
| Individual | 0.634 | 0.147 | 0.282 | 0.226 | 0.322 |
| TA | 0.560 | 0.111 | 0.082 | 0.085 | 0.209 |
| FDAs (TA, W) | **0.602** | 0.124 | 0.098 | 0.079 | 0.226 |
| FDAs (TA, G) | 0.600 | **0.126** | **0.100** | **0.098** | **0.231** |

Table 3: Performance of merging LLama2-13b-Alpaca and WizardMath-13B on Code and Math tasks. "W" denotes the weight initialization; "G" denotes the Gaussian initialization.

## 5  ABLATION STUDY

We investigate the impact of different construction choices on the quality of the FDAs. The quality is defined by the average multi-task performance of models from Eq. 5, with higher performance indicating better-quality FDAs. Experimental details are provided in Appendix D.

### 5.1  THE INITIALIZATION SCHEME

We evaluate effects of initialization schemes on previous eight ViT-B/32 checkpoints. For Gaussian initialization, we consider: $\sigma = 10^1, 10^0, 10^{-2}, 10^{-4}$. As shown in Table 4, initialization significantly affects the quality of FDAs. As $\sigma$ decreases from $10^1$, the performance first increases and then

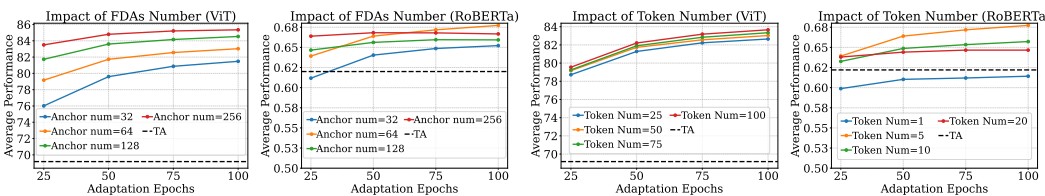

Figure 8: Multi-task performance of FDAs with different shape of FDAs on ViT-B/32 and RoBERTa-base.

decreases, consistent with our analysis. FDAs by weight perform best, aligning with their lowest optimization loss (Fig. 4). Despite a wide range of settings, FDAs consistently outperform TA.

## 5.2 THE SHAPE OF FDAS

We study the impact of the number of anchors (anchor_num) and tokens (token_num) on the quality of FDAs. We vary anchor_num over $\{32, 64, 128, 256\}$ and token_num over $\{25, 50, 75, 100\}$ for ViT-B/32 and $\{1, 5, 10, 20\}$ for RoBERTa-Base, evaluating FDAs across their respective datasets. Performance at different adaptation epochs is also reported. From Fig. 8, larger FDAs generally lead to better quality, as reflected in the multi-task performance. This is reasonable as larger optimization space makes it easier to reach a lower loss. However, for RoBERTa-Base, the average performance decreases when token_num increases from 5 to 20. Further related analysis is in Appendix D.

| Init. Scheme | ViT-B/32 |
|---|---|
| $\sigma = 10^1$ | 77.42 |
| $\sigma = 10^0$ | 81.78 |
| $\sigma = 10^{-2}$ | 83.03 |
| $\sigma = 10^{-4}$ | 71.15 |
| Weight | 83.75 |

Table 4: Multi-task performance of FDAs with different initialization schemes.

## 5.3 THE EFFECT OF DISTANCE FUNCTIONS

Distance function Dist influences both the construction (Eq. 1) and the adaptation (Eq. 5, 6). We evaluate three metrics, cosine, $\ell_1$, and $\ell_2$, for their impacts on FDAs. From Fig. 9, Dist matters more during construction than adaptation. Overall, cosine distance constructs the highest-quality FDAs, $\ell_1$ performs the worst, and our method consistently outperforms TA across all metrics.

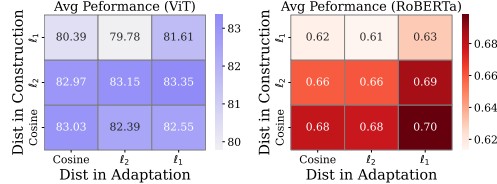

Figure 9: Multi-task performance of FDAs with different Dist functions on ViT-B/32 and RoBERTa-base.

## 5.4 OPTIMIZATION STEPS IN CONSTRUCTION

We observe the effect of the number of optimization steps on FDAs. From Fig. 10, more steps consistently improve their quality. For ViT-B/32, high-quality FDAs can be obtained from random noise in as few as 40 steps, indicating that our optimization is efficient.

## 6 RELATED WORK AND CONCLUDING REMARKS

**Related work.** Recently, the prevailing paradigm in model merging is the scaled addition of task vectors (Ilharco et al., 2022). This paradigm offers a perspective that knowledge could be transferred through parameters. Motivated by this insight, diverse parameter-centric methods for model merging have emerged. One line of works exploit the structural priors in the parameter space and adjust the task vectors (Yadav et al., 2023; Yu et al., 2024; Davari & Belilovsky, 2024; Zheng & Wang, 2024; Wei et al., 2025b; Xiong et al., 2024; Gargiulo et al., 2025; Cheng et al., 2025). In parallel, another line of works tries to introduce the data-driven priors to guide the adjustments (Matena & Raffel, 2022; Jin et al., 2022; Yang et al., 2023; 2024; Wei et al., 2025a; Xu et al., 2025; Yang et al., 2025). The unifying characteristic of both approaches is their emphasis on modeling the parameter space. Instead of modeling in the parameter space, FDAs encode the task-specific knowledge in the input-representation space, which provides an alternative perspective on model merging by extending input-space modeling.

**Concluding remarks.** This paper introduces a novel input-space-centric model merging framework. The obtained synthetic data (FDAs) models the task-specific knowledge in the parameters through their induced gradient. FDAs can be used independently or alongside existing parameter-centric methods. Experiments demonstrate the effectiveness of FDAs in model merging.

Figure 10: Multi-task performance of FDAs with different optimization steps.

## ETHICS STATEMENT

This work adheres to the ICLR Code of Ethics. In this study, no human subjects or animal experimentation was involved. All datasets used were sourced in compliance with relevant usage guidelines, ensuring no violation of privacy. We have taken care to avoid any biases or discriminatory outcomes in our research process. No personally identifiable information was used, and no experiments were conducted that could raise privacy or security concerns. We are committed to maintaining transparency and integrity throughout the research process.

## REPRODUCIBILITY STATEMENT

We provide detailed hyperparameter settings in Section 4 and Appendix B. To further facilitate reproducibility, we will release our implementation and trained model checkpoints, enabling the reported results to be reproduced.

## THE USE OF LARGE LANGUAGE MODELS (LLMS)

We used large language models (LLMs) to aid in polishing the writing. Specifically, LLMs were employed to improve grammar, clarity, and readability of the manuscript. No part of the research ideation, methodological design, or experimental analysis relied on LLMs.

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

# A   APPENDIX

## A.1   THE DERIVATION FOR EQUATION 3

We first recall the settings. Given a linear encoder $\varphi$, *i.e.*, $\boldsymbol{y} = \boldsymbol{W}\boldsymbol{x}$ with $\boldsymbol{W} \in \mathbb{R}^{d \times d}, \boldsymbol{x} \in \mathbb{R}^d$, the corresponding pretrained parameter and the downstream parameter on the $i$-th task are denoted by $\boldsymbol{W}_0$ and $\boldsymbol{W}_i$, respectively. Assuming that $\mathrm{Dist}(\boldsymbol{W}_0\boldsymbol{x}, \boldsymbol{W}_i\boldsymbol{x}) = \frac{1}{2}\|\boldsymbol{W}_0\boldsymbol{x} - \boldsymbol{W}_i\boldsymbol{x}\|_2^2$, we derive the iteration formula via gradient descent.

*Proof.* The objective function can be written as follows:

$$\min_{\boldsymbol{x}} \ \mathrm{cos\_dist}\left(\nabla_{\boldsymbol{W}} \tfrac{1}{2}\big\|\boldsymbol{W}\boldsymbol{x} - \boldsymbol{W}_i\boldsymbol{x}\big\|_2^2\Big|_{\boldsymbol{W}=\boldsymbol{W}_0}, \ \boldsymbol{W}_i - \boldsymbol{W}_0\right). \tag{7}$$

Let $\eta > 0$ be the step size and $t \in \{0, 1, 2, \dots\}$ the iteration index. The gradient-descent update is

$$\boldsymbol{x}_{t+1} = \boldsymbol{x}_t - \eta \, \nabla_{\boldsymbol{x}_t} \mathrm{cos\_dist}\left(\nabla_{\boldsymbol{W}} \tfrac{1}{2}\big\|\boldsymbol{W}\boldsymbol{x}_t - \boldsymbol{W}_i\boldsymbol{x}_t\big\|_2^2\Big|_{\boldsymbol{W}=\boldsymbol{W}_0}, \ \boldsymbol{W}_i - \boldsymbol{W}_0\right). \tag{8}$$

Since $\mathrm{cos\_dist}(A, B) = 1 - \frac{\langle A, B\rangle_F}{\|A\|_F \|B\|_F}$ with $\langle A, B\rangle_F := \mathrm{tr}(A^\top B)$, we rewrite equation 8 as

$$\boldsymbol{x}_{t+1} = \boldsymbol{x}_t + \eta \, \Delta_t, \qquad \Delta_t := \nabla_{\boldsymbol{x}_t} \frac{\left\langle \nabla_{\boldsymbol{W}} \tfrac{1}{2}\big\|\boldsymbol{W}\boldsymbol{x}_t - \boldsymbol{W}_i\boldsymbol{x}_t\big\|_2^2\Big|_{\boldsymbol{W}=\boldsymbol{W}_0}, \ \boldsymbol{W}_i - \boldsymbol{W}_0\right\rangle_F}{\left\|\nabla_{\boldsymbol{W}} \tfrac{1}{2}\big\|\boldsymbol{W}\boldsymbol{x}_t - \boldsymbol{W}_i\boldsymbol{x}_t\big\|_2^2\Big|_{\boldsymbol{W}=\boldsymbol{W}_0}\right\|_F \|\boldsymbol{W}_i - \boldsymbol{W}_0\|_F}. \tag{9}$$

**Step 1: Computing the $\boldsymbol{W}$-gradient.** For the $j$-th row $\boldsymbol{W}_{j,:}$,

$$\nabla_{\boldsymbol{W}_{j,:}} \tfrac{1}{2}\big\|\boldsymbol{W}\boldsymbol{x}_t - \boldsymbol{W}_i\boldsymbol{x}_t\big\|_2^2\Big|_{\boldsymbol{W}=\boldsymbol{W}_0} = \big(\boldsymbol{W}_0\boldsymbol{x}_t - \boldsymbol{W}_i\boldsymbol{x}_t\big)_j \, \boldsymbol{x}_t^\top. \tag{10}$$

Stacking rows gives

$$\nabla_{\boldsymbol{W}} \tfrac{1}{2}\big\|\boldsymbol{W}\boldsymbol{x}_t - \boldsymbol{W}_i\boldsymbol{x}_t\big\|_2^2\Big|_{\boldsymbol{W}=\boldsymbol{W}_0} = \big(\boldsymbol{W}_0 - \boldsymbol{W}_i\big) \boldsymbol{x}_t\boldsymbol{x}_t^\top = -\Delta\boldsymbol{W} \, \boldsymbol{x}_t\boldsymbol{x}_t^\top, \qquad \Delta\boldsymbol{W} := \boldsymbol{W}_i - \boldsymbol{W}_0. \tag{11}$$

**Step 2: Plugging equation 11 into $\Delta_t$.** The numerator in equation 9 becomes

$$\big\langle -\Delta\boldsymbol{W} \, \boldsymbol{x}_t\boldsymbol{x}_t^\top, \ \Delta\boldsymbol{W}\big\rangle_F = -\mathrm{tr}\big(\boldsymbol{x}_t\boldsymbol{x}_t^\top \Delta\boldsymbol{W}^\top \Delta\boldsymbol{W}\big) = -\boldsymbol{x}_t^\top \Delta\boldsymbol{W}^\top \Delta\boldsymbol{W} \, \boldsymbol{x}_t = -\|\Delta\boldsymbol{W}\boldsymbol{x}_t\|_2^2. \tag{12}$$

The denominator equals

$$\big\|-\Delta\boldsymbol{W} \, \boldsymbol{x}_t\boldsymbol{x}_t^\top\big\|_F \|\Delta\boldsymbol{W}\|_F = \|\Delta\boldsymbol{W}\boldsymbol{x}_t\|_2 \|\boldsymbol{x}_t\|_2 \|\Delta\boldsymbol{W}\|_F. \tag{13}$$

Hence the scalar inside the gradient is

$$-\frac{\|\Delta\boldsymbol{W}\boldsymbol{x}_t\|_2}{\|\boldsymbol{x}_t\|_2 \|\Delta\boldsymbol{W}\|_F}. \tag{14}$$

Therefore,

$$\Delta_t = \nabla_{\boldsymbol{x}_t}\left(-\frac{\|\Delta\boldsymbol{W}\boldsymbol{x}_t\|_2}{\|\boldsymbol{x}_t\|_2 \|\Delta\boldsymbol{W}\|_F}\right). \tag{15}$$

**Step 3: Evaluating $\Delta_t$ (assume $\boldsymbol{x}_t \neq 0$ and $\Delta\boldsymbol{W}\boldsymbol{x}_t \neq 0$).** Using $\nabla_{\boldsymbol{u}}\|A\boldsymbol{u}\|_2 = \frac{A^\top A\boldsymbol{u}}{\|A\boldsymbol{u}\|_2}$ and $\nabla_{\boldsymbol{u}}\|\boldsymbol{u}\|_2 = \frac{\boldsymbol{u}}{\|\boldsymbol{u}\|_2}$,

$$\Delta_t = -\frac{1}{\|\Delta\boldsymbol{W}\|_F}\left[\frac{\Delta\boldsymbol{W}^\top \Delta\boldsymbol{W} \, \boldsymbol{x}_t}{\|\Delta\boldsymbol{W}\boldsymbol{x}_t\|_2 \|\boldsymbol{x}_t\|_2} - \frac{\|\Delta\boldsymbol{W}\boldsymbol{x}_t\|_2}{\|\boldsymbol{x}_t\|_2^3} \, \boldsymbol{x}_t\right]. \tag{16}$$

**Step 4: Iteration in affine form.** Write

$$\Delta_t = \sigma_t \, \boldsymbol{x}_t + \beta_t \, \Delta\boldsymbol{W}^\top \Delta\boldsymbol{W} \, \boldsymbol{x}_t, \tag{17}$$

where

$$\sigma_t = \frac{\|\Delta \boldsymbol{W} \boldsymbol{x}_t\|_2}{\|\Delta \boldsymbol{W}\|_F \|\boldsymbol{x}_t\|_2^3} > 0, \qquad \beta_t = -\frac{1}{\|\Delta \boldsymbol{W}\|_F \|\Delta \boldsymbol{W} \boldsymbol{x}_t\|_2 \|\boldsymbol{x}_t\|_2} < 0.$$

Hence

$$\boldsymbol{x}_{t+1} = \boldsymbol{x}_t + \eta \Delta_t = (1 + \eta \sigma_t) \boldsymbol{x}_t + \eta \beta_t \Delta \boldsymbol{W}^\top \Delta \boldsymbol{W} \boldsymbol{x}_t. \tag{18}$$

Note that $\eta \sigma_t$ generally is a small positive number. Thus, for better analysis, we approximate the iteration as

$$\boldsymbol{x}_{t+1} = \boldsymbol{x}_t + \eta \Delta_t = \boldsymbol{x}_t + \eta \beta_t \Delta \boldsymbol{W}^\top \Delta \boldsymbol{W} \boldsymbol{x}_t. \tag{19}$$

$\square$

## A.2 THE PROOF FOR PROPOSITION 2.1

*Proof of Proposition 2.1.* We start from the iteration in equation 3:

$$\boldsymbol{x}_{t+1} = \boldsymbol{x}_t + \eta \beta_t \Delta \boldsymbol{W}^\top \Delta \boldsymbol{W} \boldsymbol{x}_t, \quad \Delta \boldsymbol{W} = \boldsymbol{W}_i - \boldsymbol{W}_0, \quad t = 0, \ldots, T-1.$$

And we have that: $\Delta \boldsymbol{W}^\top \Delta \boldsymbol{W} = \boldsymbol{U} \boldsymbol{\Lambda} \boldsymbol{U}^\top$, where $\boldsymbol{U} = [\boldsymbol{u}_1, \ldots, \boldsymbol{u}_d] \in \mathbb{R}^{d \times d}$ is orthogonal and $\boldsymbol{\Lambda} = \mathrm{diag}(\lambda_1, \ldots, \lambda_d)$ with $\lambda_1 > \cdots > \lambda_d > 0$.

**Step 1: Project $\boldsymbol{x}_t$ onto the eigenbasis.** Let

$$\boldsymbol{x}_t = \sum_{i=1}^d c_t^i \boldsymbol{u}_i,$$

where $c_t^i = \boldsymbol{u}_i^\top \boldsymbol{x}_t$. Then

$$\Delta \boldsymbol{W}^\top \Delta \boldsymbol{W} \boldsymbol{x}_t = \Delta \boldsymbol{W}^\top \Delta \boldsymbol{W} \sum_{i=1}^d c_t^i \boldsymbol{u}_i = \sum_{i=1}^d c_t^i \lambda_i \boldsymbol{u}_i.$$

**Step 2: Plugging the projection into equation 3.**

$$\boldsymbol{x}_{t+1} = \boldsymbol{x}_t + \eta \beta_t \sum_{i=1}^d c_t^i \lambda_i \boldsymbol{u}_i = \sum_{i=1}^d c_t^i \boldsymbol{u}_i + \sum_{i=1}^d (\eta \beta_t \lambda_i c_t^i) \boldsymbol{u}_i = \sum_{i=1}^d c_t^i (1 + \eta \beta_t \lambda_i) \boldsymbol{u}_i.$$

Define $\gamma_t = -\eta \beta_t > 0$, then $c_{t+1}^i = c_t^i (1 - \gamma_t \lambda_i)$, $\quad t = 0, \ldots, T-1$.

**Step 3: Recursion.** By recursion, we can get the formula in proposition 2.1:

$$c_t^i = c_0^i \prod_{j=0}^{t-1} (1 - \gamma_j \lambda_i), \quad i = 1, \ldots, d.$$

$\square$

## B EXPERIMENT DETAILS

### B.1 DETAILS OF DOWNSTREAM MODELS FOR MERGING

For *vision* task, we follow the setup in previous works (Ilharco et al., 2022; Yadav et al., 2023). Specifically, we adopt eight public, domain-specific foundation models from Ilharco et al. (2022), which obtained by finetuning the pretrained Vision Encoder of CLIP (Radford et al., 2021) on the following datasets: SUN397 (Xiao et al., 2016), Cars (Krause et al., 2013), RESISC45 (Cheng et al., 2017), EuroSAT (Helber et al., 2019), SVHN (Netzer et al., 2011), GTSRB (Stallkamp et al., 2011), MNIST (LeCun et al., 1998) and DTD (Cimpoi et al., 2014). All sizes of these models, i.e., ViT-B/32, ViT-B/16 and ViT-L/14, are taken into consideration.

For natural language processing task, we also follow previous works (Yu et al., 2024; Xu et al., 2025). Specifically, we consider the downstream models of eight datasets from the GLUE benchmark (Wang et al., 2018a), including CoLA (Warstadt et al., 2018), SST-2 (Socher et al., 2013),

MRPC (Dolan & Brockett, 2005), STS-B (Cer et al., 2017), QQP (Iyer et al., 2017), MNLI (Williams et al., 2017), QNLI (Wang et al., 2018a; Rajpurkar et al., 2016), RTE (Wang et al., 2018a; Haim et al., 2006; Giampiccolo et al., 2007; Bentivogli et al., 2009). To obtain the downstream models, we follow the finetuning procedure of Yu et al. (2024) on publicly available pretrained RoBERTa-base and RoBERTa-large.

For auto-regressive models, we follow the practice in Yu et al. (2024) and consider two expert models: the Math expert model WizardMath-13B (Luo et al., 2023) and the Code expert model LLaMA-2-13B-Code-Alpaca. We use four datasets for evaluation, including GSM8K (Cobbe et al., 2021), MATH (Hendrycks et al., 2020), HumanEval (Chen et al., 2021) and MBPP (Austin et al., 2021). For evaluation, we adopt the evaluation codes of Xu et al. (2025).

## B.2 DETAILS OF BASELINE METHODS

For data-based baselines (*i.e.*, RegMean (Jin et al., 2022), Fisher merging (Matena & Raffel, 2022), AdaMerging (Yang et al., 2023), and ProDistill (Xu et al., 2025)), we follow the released implementations. For ViT, we adopt the results reported in Xu et al. (2025) as they rely on the same public checkpoints. For LLM, we simply set the coefficient of TA method as $0.5$, which is adopted in Xu et al. (2025). For data-free baseline methods (*i.e.*, TA (Ilharco et al., 2022), TSV (Gargiulo et al., 2025), and WUDI (Cheng et al., 2025)), we use their publicly available open-source code. We try our best to ensure fair and strong baselines.

## B.3 DETAILS OF FDAS

All FDAs in our experiments follows the layer-wise manner. We keep the settings of the construction and adaptation consistent across layers. Both Gaussian and parameter initializations are considered. For Gaussian initialization, we set $\sigma = 0.01$. Both initializations share the same settings.

**FDA Construction.** For ViT and RoBERTa, we first set the number $n$ of anchors as $64$ for each task. Then, the token_num of FDAs for ViT follows the default settings: $50$ for ViT-B/32, $197$ for ViT-B/16, and $257$ for ViT-L/14. For RoBERTa-base and RoBERTa-large, we fix the token_num as $5$. To optimize these anchors, we adopt the AdamW optimizer (Loshchilov & Hutter, 2017), iterating for 1200 steps with a learning rate of $1e-2$. For WizardMATH and llama-2-13b-alpaca, we construct FDAs for feed-forword networks. Thus, we only set the number $n$ of anchors as $8192$. We also adopt AdamW optimizer, iterating for 200 steps with a learning rate of $1e-2$. All the above optimizations are performed with a full batch size.

**Parameter Optimization.** We adopt the Adam optimizer to optimize parameters. There are three hyperparameters: learning rate, batch size and optimization epochs. For FDAs from different initialization schemes, we adopt the same settings. When the initial model is initialized by pretrained parameter, we adopt Eq. 5. We use a batch size of 128 for all models. For all ViT models, we set the learning rate to 1e-5 and train for 100 epochs. For all RoBERTa, we use a learning rate of 5e-5, training for 100 epochs on the base model and 50 epochs on the large model. When the initial model is initialized by task-vector-based merging method, we adopt Eq. 6. We follow previous works (Yang et al., 2023; Xu et al., 2025) and use a large learning rate $1e-2$. Generally, for ViT and RoBERTa, we use a batch size of 128, also training for 100 epochs (15 epochs for RoBERTa-large). For the initial ViT model by WUDI, we set the batch size as 512 and train for 25 epochs. For the auto-regressive model, we use a batch size of 8192, training for 25 epochs.

## B.4 REMAINING RESULTS.

We present the experimental results on ViT-B/32, ViT-B/L-14 and RoBERTa-base on Table 5, 6 and 7, respectively. FDAs bring slight improvements on the WUDI-initialized merged model. Please note that WUDI-initialzed model has already achieves $98.3\%$ of the performance of eight individual models. That means that this initialization is already situated in a well-optimized local minima. Generally, these results are consistent with the observations in our main paper.

| Method | SUN397 | Cars | RESISC45 | EuroSAT | SVHN | GTSRB | MNIST | DTD | Avg |
|---|---|---|---|---|---|---|---|---|---|
| Individual | 75.34 | 77.73 | 95.98 | 99.89 | 97.46 | 98.73 | 99.69 | 79.36 | 90.52 |
| RegMean | 67.47 | 66.63 | 81.75 | 93.33 | 86.68 | 79.92 | 97.30 | 60.16 | 79.15 |
| Fisher merging | 63.95 | 63.84 | 66.86 | 83.48 | 79.54 | 60.11 | 91.27 | 49.36 | 69.80 |
| AdaMerging | 63.69 | 65.74 | 77.65 | 91.00 | 82.48 | 93.12 | 98.27 | 62.29 | 79.28 |
| Representation Surgery | 71.20 | 72.00 | 92.30 | 99.00 | 92.20 | 97.90 | 99.00 | 76.10 | 87.50 |
| ProDistill | 68.90 | 71.21 | 89.89 | 99.37 | 96.13 | 95.29 | 99.46 | 68.03 | 86.04 |
| TA | 55.16 | 54.98 | 66.68 | 78.89 | 80.21 | 69.68 | 97.34 | 50.37 | 69.16 |
| TSV-M | 69.08 | 70.92 | 85.67 | 95.15 | 92.02 | 91.93 | 99.25 | 69.20 | 84.15 |
| WUDI | 70.92 | 71.38 | 85.68 | 96.33 | 94.27 | 94.51 | 99.47 | 69.47 | 85.26 |
| FDAs (Pretrained, Gauss) | 67.46 | 69.05 | 81.87 | 96.89 | 94.02 | 89.58 | 99.28 | 66.12 | 83.03 |
| FDAs (Pretrained, Weight) | 68.12 | 70.46 | 83.94 | 97.07 | 94.08 | 90.03 | 99.33 | 66.97 | 83.75 |
| FDAs (TA, Gauss) | 69.48 | 71.43 | 83.79 | 96.89 | 94.43 | 91.20 | 99.36 | 68.67 | 84.41 |
| FDAs (TA, Weight) | 69.61 | 71.83 | 85.27 | 97.00 | 94.33 | 91.59 | 99.39 | 69.10 | 84.76 |
| FDA (TSV, Gauss) | 71.17 | 73.25 | 86.46 | 91.19 | 94.25 | 92.03 | 99.39 | 70.64 | 85.55 |
| FDA (TSV, Weight) | 71.23 | 73.71 | 86.76 | 97.19 | 94.11 | 91.79 | 99.44 | 70.74 | 85.62 |
| FDA (WUDI, Gauss) | 72.71 | 73.71 | 86.97 | 96.67 | 94.20 | 93.99 | 99.42 | 70.32 | 86.00 |
| FDA (WUDI, Weight) | 72.82 | 73.88 | 87.02 | 96.70 | 94.13 | 93.76 | 99.40 | 70.59 | 86.04 |

Table 5: Performance of merging ViT-B/32 models across eight downstream vision tasks. FDAs (*init model*, *FDAs init*) denotes the choice of the initial model and the initialization strategies for FDAs, respectively. "Pretrained" denotes the initial model is the pretrained model.

| Method | SUN397 | Cars | RESISC45 | EuroSAT | SVHN | GTSRB | MNIST | DTD | Avg |
|---|---|---|---|---|---|---|---|---|---|
| Individual | 82.32 | 92.35 | 97.38 | 99.78 | 98.11 | 99.24 | 99.69 | 84.15 | 94.13 |
| RegMean | 74.04 | 87.22 | 88.52 | 98.15 | 92.89 | 90.22 | 99.27 | 69.84 | 87.52 |
| Fisher merging | 71.28 | 85.18 | 81.59 | 89.67 | 81.51 | 83.39 | 96.31 | 65.48 | 81.80 |
| AdaMerging | 75.96 | 89.42 | 90.08 | 96.59 | 91.78 | 97.52 | 98.91 | 77.61 | 89.73 |
| Representation Surgery | 80.30 | 90.80 | 94.30 | 98.20 | 94.10 | 98.70 | 99.20 | 82.50 | 92.30 |
| ProDistill | 77.73 | 90.04 | 94.43 | 99.48 | 97.71 | 98.26 | 99.63 | 78.24 | 91.94 |
| TA | 74.16 | 82.09 | 86.67 | 94.07 | 87.91 | 86.77 | 98.94 | 65.69 | 84.54 |
| TSV | 79.00 | 89.99 | 93.95 | 99.15 | 95.34 | 96.16 | 99.51 | 79.10 | 91.52 |
| WUDI | 81.15 | 90.95 | 94.00 | 99.33 | 96.21 | 98.04 | 99.63 | 80.64 | 92.49 |
| FDAs (Pretrained, Gauss) | 77.59 | 90.05 | 92.75 | 99.04 | 95.42 | 96.78 | 99.56 | 76.76 | 90.99 |
| FDAs (Pretrained, Weight) | 77.91 | 90.14 | 92.84 | 99.04 | 95.44 | 96.56 | 99.59 | 76.86 | 91.05 |
| FDAs (TA, Gauss) | 78.96 | 90.41 | 93.13 | 99.07 | 95.63 | 97.15 | 99.58 | 77.23 | 91.40 |
| FDAs (TA, Weight) | 78.92 | 90.35 | 93.19 | 99.11 | 95.53 | 96.83 | 99.61 | 77.13 | 91.33 |
| FDAs (TSV, Gauss) | 79.84 | 90.66 | 93.95 | 99.19 | 95.81 | 97.35 | 99.61 | 79.04 | 91.93 |
| FDAs (TSV, Weight) | 79.69 | 90.60 | 93.68 | 99.19 | 95.51 | 97.05 | 99.60 | 78.40 | 91.71 |
| FDAs (WUDI, Gauss) | 81.09 | 91.16 | 94.41 | 99.26 | 96.21 | 97.86 | 99.68 | 80.37 | 92.51 |
| FDAs (WUDI, Weight) | 81.07 | 91.27 | 94.48 | 99.26 | 96.11 | 97.72 | 99.67 | 80.05 | 92.45 |

Table 6: Performance of merging ViT-L/14 models across eight downstream vision tasks. FDAs (*init model*, *FDAs init*) denotes the choice of the initial model and the initialization strategies for FDAs, respectively. "Pretrained" denotes the initial model is the pretrained model.

### B.5 A PRACTICAL METHOD FOR CHOOSING THE SCALING COEFFICIENT

As discussed in the main paper, the scaling coefficient $\sigma$ is crucial for the convergence. We provide a practical heuristic to choose $\sigma$. Specifically, we fix one layer with the same initial FDAs and evaluate a set of candidate $\sigma$ values by comparing their alignment after a fixed number of iterations, selecting the $\sigma$ that yields the best alignment as the scaling coefficient.

## C EXTENSIONS FOR SEC. 3

In this section, we describe the details in the investigation for the knowledge in FDAs, and then present further examples.

**Details of Observation 1.** We first follow the same construction procedure in Sec. B.3 and obtain sets of FDAs $\{x_{ij}^{(l)}\}_{j=1}^{64}, x_{ij} \in \mathbb{R}^{50 \times 768}, i = 1, \ldots, 8; l = 1, \ldots, 12$. Then we unfold each set into the matrix $X_i^{(l)} \in \mathbb{R}^{3200 \times 768}$, treating each token embedding as a representation unit (Clark et al., 2019; Dosovitskiy et al., 2020; Raghu et al., 2021). We conduct the singular value decomposition for $X_i^{(l)}$ and obtain the sorted singular values: $\lambda_{i,1}^{(l)}, \ldots, \lambda_{i,768}^{(l)}$. The normalized singular values $\tilde{\lambda}_{ij}^{(l)}$ are computed as $\tilde{\lambda}_{ij}^{(l)} = \lambda_{ij}^{(l)} / \lambda_{i,1}^{(l)}$. We visualize more results in Fig. 11.

**Details of Observation 2.** For the features of real task-specific data in the $l$-th layer, we attach hooks at the corresponding layers of the downstream checkpoints to extract features. 64 real examples are

| Method | CoLA | SST-2 | MRPC | STS-B | QQP | MNLI | QNLI | RTE | Avg |
|---|---|---|---|---|---|---|---|---|---|
| Individual | 0.626 | 0.9427 | 0.8946 | 0.9070 | 0.8986 | 0.8721 | 0.9257 | 0.7581 | 0.8531 |
| RegMean | 0.2078 | 0.9266 | 0.8215 | 0.5350 | 0.8141 | 0.7551 | 0.8541 | 0.7256 | 0.7050 |
| Fisher merging | 0.123 | 0.8188 | 0.7598 | -0.1194 | 0.7319 | 0.6056 | 0.507 | 0.4874 | 0.4893 |
| AdaMerging | 0.0864 | 0.8968 | 0.795 | 0.398 | 0.7936 | 0.7579 | 0.7128 | 0.7076 | 0.6435 |
| ProDistill | 0.4968 | 0.9209 | 0.8340 | 0.6623 | 0.8044 | 0.7987 | 0.8918 | 0.7148 | 0.7655 |
| TA | 0.0257 | 0.9048 | 0.7916 | 0.2873 | 0.8169 | 0.7437 | 0.7216 | 0.7220 | 0.6267 |
| TSV | 0.0722 | 0.9014 | 0.806 | 0.3081 | 0.8365 | 0.8031 | 0.7893 | 0.7401 | 0.6571 |
| WUDI | 0.1459 | 0.922 | 0.7925 | 0.3832 | 0.8393 | 0.7917 | 0.7972 | 0.7292 | 0.6751 |
| FDAs (Pretrained, Gauss) | 0.2178 | 0.9232 | 0.8144 | 0.4256 | 0.8019 | 0.7065 | 0.7928 | 0.7365 | 0.6773 |
| FDAs (Pretrained, Weight) | 0.2229 | 0.9209 | 0.8057 | 0.2291 | 0.8117 | 0.6871 | 0.8294 | 0.7329 | 0.6550 |
| FDAs (TA, Gauss) | 0.2304 | 0.9174 | 0.8124 | 0.6029 | 0.7763 | 0.7679 | 0.7366 | 0.6968 | 0.6926 |
| FDAs (TA, Weight) | 0.2119 | 0.9083 | 0.8215 | 0.5445 | 0.7929 | 0.75 | 0.7424 | 0.7112 | 0.6853 |
| FDAs (TSV, Gauss) | 0.1923 | 0.8865 | 0.7962 | 0.4612 | 0.8064 | 0.7796 | 0.7695 | 0.6895 | 0.6727 |
| FDAs (TSV, Weight) | 0.2604 | 0.8979 | 0.8200 | 0.2487 | 0.8231 | 0.7930 | 0.8052 | 0.7256 | 0.6717 |
| FDAs (WUDI, Gauss) | 0.2231 | 0.9278 | 0.7838 | 0.3999 | 0.8218 | 0.8015 | 0.7926 | 0.7292 | 0.6850 |
| FDAs (WUDI, Weight) | 0.2872 | 0.9289 | 0.8108 | 0.3450 | 0.8263 | 0.7992 | 0.8038 | 0.7292 | 0.6913 |

Table 7: Performance of merging RoBERTa-base models across eight NLU tasks.

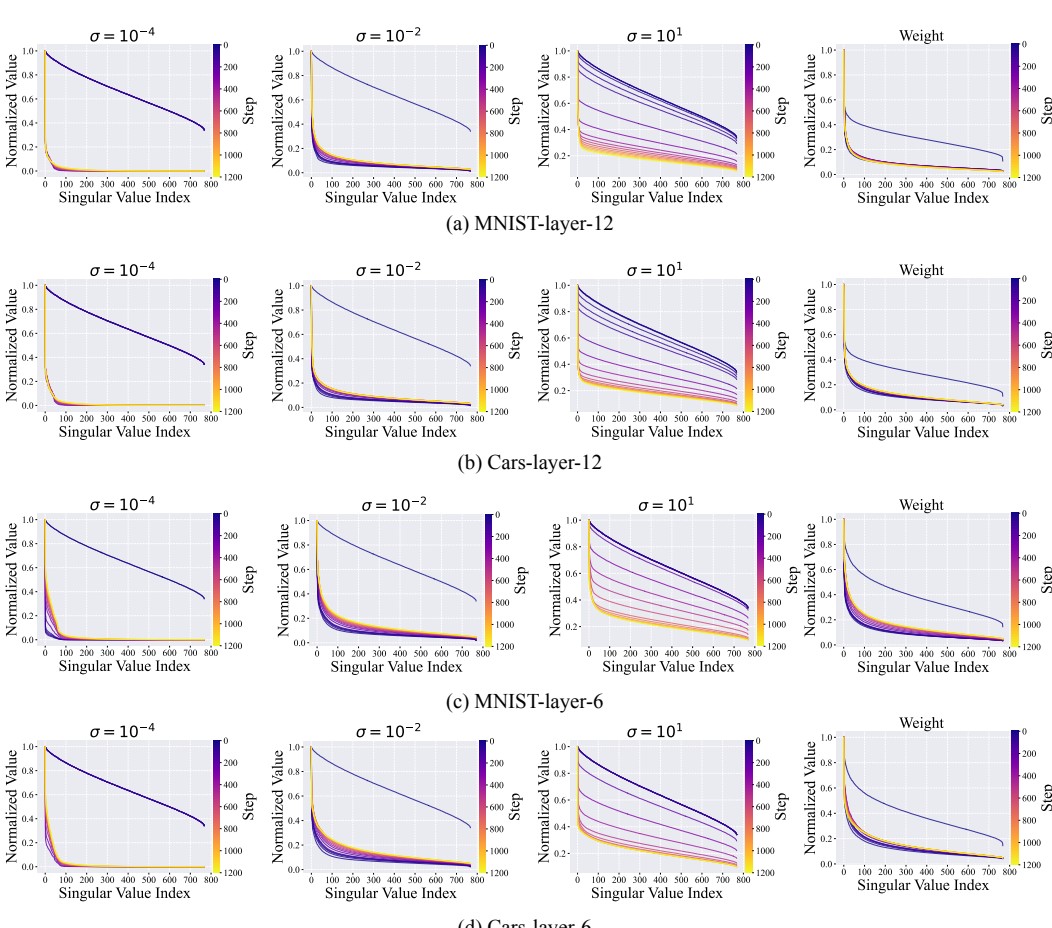

(a) MNIST-layer-12

(b) Cars-layer-12

(c) MNIST-layer-6

(d) Cars-layer-6

Figure 11: Evolution of Normalized singular values of FDAs in the FDA construction. $\sigma = 10^1$ denotes FDAs initialized by sampling from $\mathcal{N}(\mathbf{0}, \mathbf{I}_d)$ and scaling by $10^1$; "Weight" denotes that of FDAs initialized from linear weight. FDAs of different initialization schemes tend to evolve into long-tailed structures.

randomly sampled from validation dataset for each task. We unfold them into the matrices denoted as $\hat{\mathbf{X}}_i^l, i = 1, \ldots, 8; l = 1, \ldots, 12$. Then we perform the projection matrix method to analyze the similarity of subspaces spanned by the top 20% singular vectors. Assume that $\mathbf{U}_i^{(l)}$ and $\hat{\mathbf{U}}_i^{(l)}$ are spanned by their top 20% singular vectors, the similarity is measured by:

$$\text{Sim} = \frac{\text{tr}(\mathbf{P}_i^l \hat{\mathbf{P}}_i^l)}{768 \times 20\%},$$

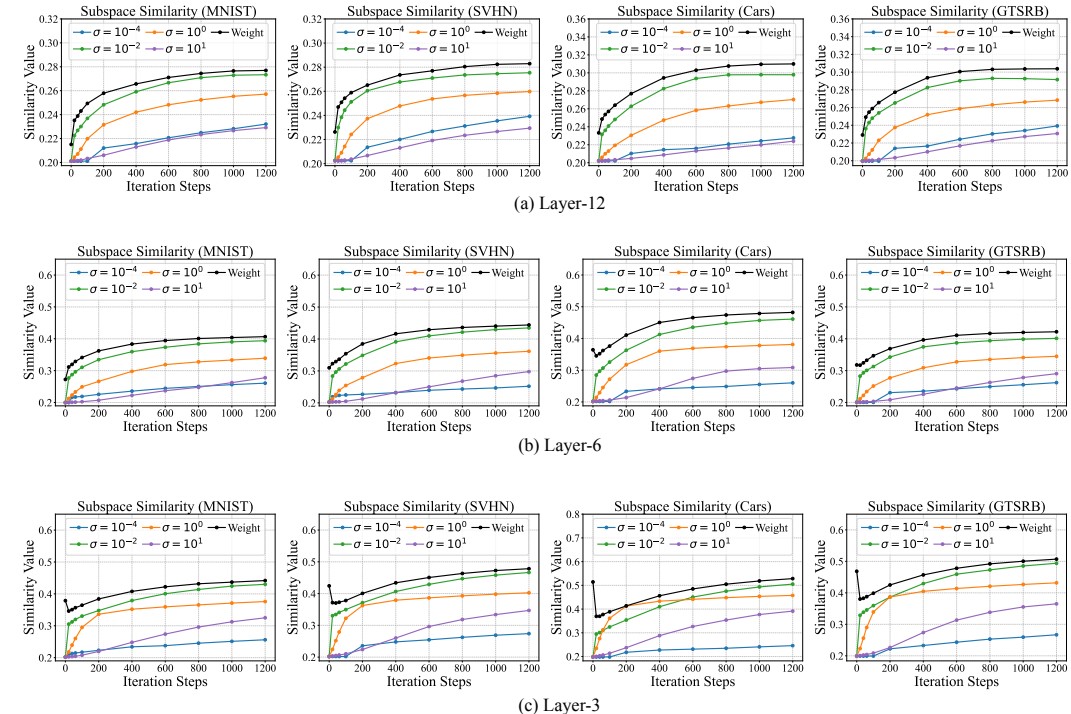

Figure 12: Evolution of subspace similarity of FDAs in the FDA construction. $\sigma = 10^1$ denotes the FDAs initialized by sampling from $\mathcal{N}(\mathbf{0}, \boldsymbol{I}_d)$ and scaling by $10^1$; "Weight" denotes the FDAs initialized from linear weight. FDAs of different initialization schemes tend to align the subspace spanned by real data.

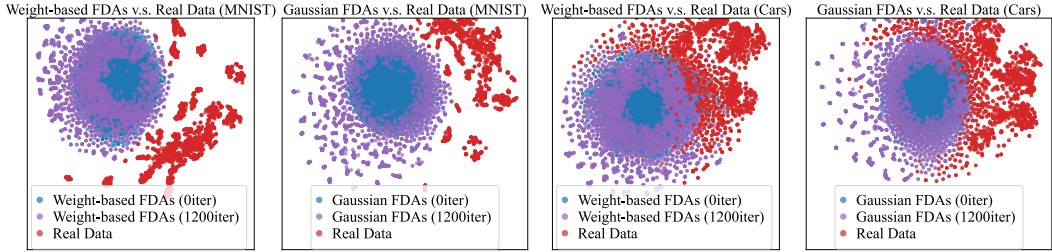

Figure 13: The low-dimensional visualization via t-SNE of FDAs and real data.

where $\boldsymbol{P}_i^l, \hat{\boldsymbol{P}}_i^l$ are the projection matrices computed by $\boldsymbol{P}_i^{(l)} = \boldsymbol{U}_i^{(l)}(\boldsymbol{U}_i^{(l)})^\top$, $\hat{\boldsymbol{P}}_i^{(l)} = (\hat{\boldsymbol{U}}_i^{(l)})(\hat{\boldsymbol{U}}_i^{(l)})^\top$. We present more results in Fig. 12.

Moreover, we adopt the t-SNE visualization to observe whether the optimization process drives FDAs closer to the manifold of real features. As shown in Fig. 13, there is no clear evidence that optimization process makes the initial anchors closer to the manifold of real data.

**Details of Observation 3.** To acquire the parameter update vectors, we follow the finetuning procedure in Ilharco et al. (2022) and sample the parameter update vectors from consecutive batches. The fine-tuning procedure is performed starting from both the pretrained model and the merged model obtained by TA. This yields two sets of updated task vectors: one initialized from the pretrained model and the other from the merged model. Instead of completing full fine-tuning, we sample 512 vectors per task and then stop. These sampled vectors are used to span the corresponding cones. We denote the sampled vectors for $i$-th task as $\Delta \boldsymbol{w}_{i,1}, \ldots, \Delta \boldsymbol{w}_{i,512} \in \mathbb{R}^p$. We use the $\boldsymbol{\tau}_i' \in \mathbb{R}^p$ to denote the adaptation direction induced by FDAs. Then, we solve the following non-negative least square problem to obtain the projection energy ratio:

$$\mathrm{Ratio}_i = \frac{\|[\Delta \boldsymbol{w}_{i,1}, \ldots, \Delta \boldsymbol{w}_{i,512}] \, \boldsymbol{\alpha}_i\|_F}{\|\boldsymbol{\tau}_i\|_F}, \quad \text{where } \boldsymbol{\alpha}_i = \arg \min_{\substack{\alpha_{ij} \geq 0 \\ j=1,\ldots,512}} \|\boldsymbol{\tau}_i - [\Delta \boldsymbol{w}_{i,1}, \ldots, \Delta \boldsymbol{w}_{i,512}] \, \boldsymbol{\alpha}_i\|_F^2 \, .$$

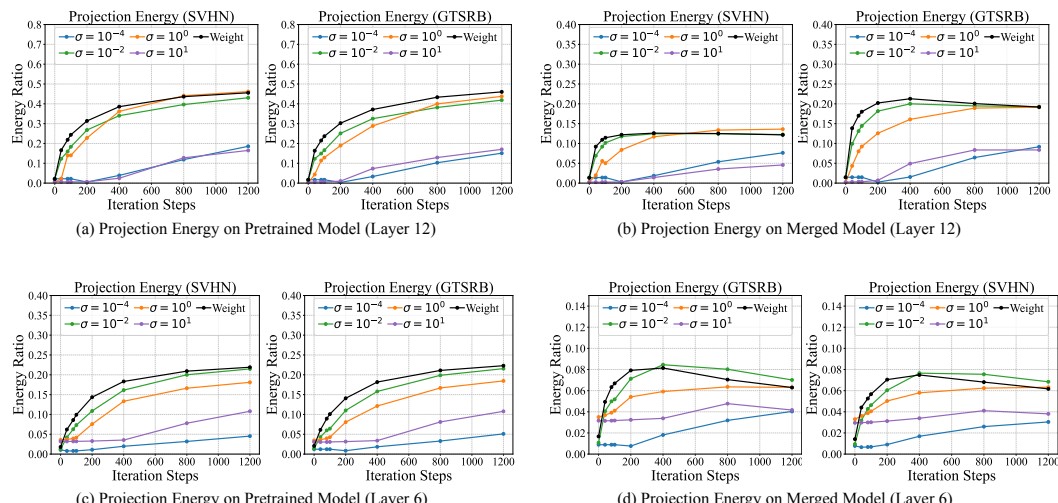

Figure 14: Evolution of projection energy ratio on pretrained model and merged model (TA). $\sigma = 10^1$ denotes the FDAs initialized by sampling from $\mathcal{N}(\mathbf{0}, \mathbf{I}_d)$ and scaling by $10^1$; "Weight" denotes the FDAs initialized from linear weight.

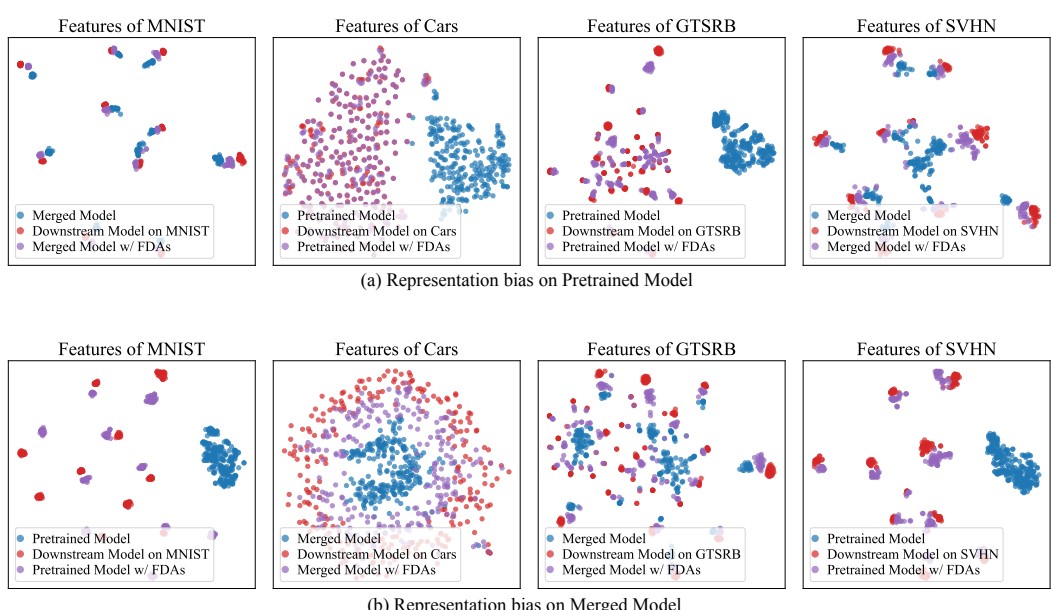

Figure 15: The Effect of FDAs in the representation. In general, the adaptation with FDAs substantially mitigates the representation bias observed in both pretrained and merged models.

Note that solving the above optimization problem, even for a single layer, is nearly intractable. Therefore, we compute it separately for the attention block and the MLP layer, and then report the averaged energy. We present more visualization of projection energy in Fig. 14 and the effects on the representation in Fig. 15. Although the improvement in shallow-layer projection energy is not significant during the optimization, FDAs still effectively alleviate the overall representation bias.

In the paper, the task vectors obtained from real data are also projected onto these cones. We take the projection energy ratio as the reference. Specifically, for the pretrained model, we directly use the task vectors corresponding to the publicly available fine-tuned checkpoints; for the merged model, we fine-tune for one epoch to obtain checkpoints, from which the task vectors are generated.

| token_num | CoLA | SST-2 | MRPC | **STSB** | QQP | MNLI | QNLI | RTE | Avg |
|-----------|------|-------|------|----------|-----|------|------|-----|-----|
| 5 | 0.2178 | 0.9232 | 0.8144 | **0.4256** | 0.8019 | 0.7065 | 0.7928 | 0.7365 | 0.6773 |
| 10 | 0.2685 | 0.9197 | 0.8043 | **0.1604** | 0.8214 | 0.7256 | 0.8215 | 0.7365 | 0.6572 |
| 20 | 0.2421 | 0.9243 | 0.8082 | **0.0904** | 0.8226 | 0.7410 | 0.8294 | 0.7148 | 0.6466 |
| 5 | 0.2020 | 0.9243 | 0.8046 | – | 0.8030 | 0.7137 | 0.8078 | 0.7112 | **0.7100** |
| 10 | 0.2432 | 0.9255 | 0.8004 | – | 0.8206 | 0.7289 | 0.8252 | 0.7184 | **0.7232** |
| 20 | 0.2468 | 0.9186 | 0.7981 | – | 0.8218 | 0.7487 | 0.8270 | 0.7148 | **0.7251** |

Table 8: Performance on each dataset of RoBERTa-Base with different token_num. The upper part includes STSB, while the lower part excludes STSB (STSB is denoted as "–").

## D    EXTENSIONS FOR SEC. 5

In this section, we first present more experimental details about ablation study. Then, we further analyze the effect of token_num in RoBERTa-base.

**Experimental Details.** To observe the effect of different initialization schemes, we follow the same construction and adaptation settings as in the experiments on ViT-B/32, while only varying the initialization scheme. For the shape of FDAs, we also follow the same construction settings and keep the batch size and learning rates in the adaptation, while varying the shape of FDAs and the adaptation steps. For the effect of distance function, we only varies the distance functions both in construction and adaptation process. For optimization steps, we only vary the optimization steps in the construction phase.

**Further analysis on** token_num. As shown in the Fig. 8, we observe that the average performance decreases when the token_num increases from 5 to 20 for RoBERTa-Base models, which appears to contradict the trends observed in ViT-B/32. We further analyze the performance of each dataset. As shown in Table 8, we find that the performance drop is mainly attributed to STSB. Therefore, we conduct merging without STSB and observe that the performance consistently increases with larger token_num, which is consistent with the trend in ViT-B/32. We hypothesize that FDAs yield better performance when their shape is closer to that of the real data space.

