# OpenReview forum: "Model Merging with Functional Dual Anchors"
_ICLR.cc/2026/Conference — Submitted to ICLR 2026_

### Official Review · Reviewer_snaV · 2025-10-22

**Soundness:** 3
**Presentation:** 2
**Contribution:** 3
**Rating:** 4
**Confidence:** 2

**Summary:**

This paper proposes a method called Functional Data Alignment (FDA) for model merging.
Instead of directly averaging or interpolating model parameters, the authors first find a set of representative input features for each layer and align the models so that their outputs remain unchanged on these features.
The intuition is that aligning model behavior in function space allows smoother merging and better preservation of learned knowledge.

**Strengths:**

Performing model merging via function alignment rather than parameter interpolation is conceptually interesting.
This paper is well-structured and easy to follow.

**Weaknesses:**

The FDA construction process relies on gradient descent, but it is unclear whether these gradients are computed per layer or over the entire network. If gradients are restricted to individual layers, the global inter-layer dependencies may be ignored. If gradients are computed for the full network, the computational cost would become prohibitively large, especially for deep models. The authors should clarify this trade-off and the precise optimization scope.

The convergence of FDA optimization is not well analyzed.
Since the gradient descent outcome depends heavily on random initialization, the authors should provide:
Results over multiple random seeds,
Mean and standard deviation of performance,
A short convergence analysis to justify robustness.
Without this, the reproducibility and reliability of FDA are questionable.

It is unclear whether the same optimization setup (learning rate, steps, loss weights) is applied uniformly across layers. Given that shallow and deep layers encode very different semantic information, a uniform optimization configuration may be inappropriate. The paper should discuss whether the FDA optimization strategy varies by layer or is shared globally, and why.

The FDA process requires layer-wise gradient-based optimization, which could be expensive for deep models such as Qwen3 or other large-scale transformers.
The paper should report or estimate. The computational time and memory cost of FDA, such as a simple simulation of time or memory cost on a large model (if you can not find fine-tuned checkpoint during rebuttal). Without such analysis, it is difficult to assess the practicality and scalability of the method.

The proposed method shows small gains in CV tasks but large improvements in NLP tasks. This discrepancy lacks explanation.

The story of merging in function space in this paper is good. However, it may be verbose and occasionally for some readers not faimilar with model merging. I think 'finding a set of representative input features for each layer and aligning the models so that their outputs remain unchanged on these features' may be a better way for understanding.

**Questions:**

See weakness.

---

> ### Author Response · Authors · 2025-11-21
> **Response to Reviewer snaV (Part 1)**
>
> ### **Q1: Gradient Computation Scope and Global Inter-layer Dependencies**
>
> Thanks for raising these important points regarding the scope of gradient computation and global inter-layer dependencies.
>
> In our default setting, FDAs are constructed independently for each layer. Specifically, for current Transformer-based deep models, FDAs are constructed for each Transformer residual block, and the model parameters are adapted on a per-layer basis accordingly.
>
> We now discuss the global inter-layer dependencies. In principle, constructing FDAs for the entire network jointly would be the optimal solution, as it fully accounts for the nonlinear interactions across all layers. However, this global strategy comes with several severe drawbacks:
>
> - substantially higher memory consumption,
> - significantly longer computation time, and
> - much more complex hyperparameter tuning.
>
> Given these practical limitations, we adopt the layer-wise strategy as a trade-off. Although this approach ignores the global nonlinear dependencies across layers, it brings crucial advantages that make FDA applicable to large Transformer models:
>
> - low memory footprint,
> - short computation time, and
> - fully parallelizable construction and adaptation.
>
> More importantly, our wide experiments show that the layer-wise FDA still yields substantial performance improvements. This empirical evidence strongly suggests that the FDA paradigm possesses considerable robustness and potential, even under the trade-off of simplified layer-wise formulation.
>
> To further examine whether the global inter-layer dependencies would overturn the conclusions drawn from our layer-wise initialization analysis, we conduct an additional experiment. Specifically, we construct FDAs jointly for two adjacent Transformer blocks and apply FDA-based optimization to these two layers simultaneously, while keeping all optimization settings unchanged. The corresponding optimization loss and multi-task performance can be found in the link:
> [Optimization Loss for Multi-layers](https://anonymous.4open.science/r/iclrsubmission884/Optimization_loss_for_Multlayers.png).
>
> The table below summarizes the multi-task performance:
>
> | Method        | $\sigma=10.0$ | $\sigma=1.0$ | $\sigma=1e-2$ | $\sigma=1e-4$ |
> |---------------|---------------|---------------|----------------|----------------|
> | FDAs (1-layer)| 77.42         | 81.78         | 83.03          | 71.75          |
> | FDAs (2-layer)| 76.78         | 81.83         | 82.6           | 70.33          |
>
> As shown in the results, constructing FDAs for multiple layers exhibits optimization dynamics and performance trends highly consistent with those of single-layer FDAs. This indicates that inter-layer dependencies have a relatively small effect on FDAs.
>
> Additionally, we observe that the performance of FDA (2-layer) is slightly lower than that of FDA (1-layer). This is reasonable because the number of anchors in FDA (2-layer) is halved compared to FDA (1-layer), and we did not perform any hyperparameter tuning for the 2-layer setting. Nevertheless, even with only half the anchors, FDA (2-layer) still retains about 99% of the performance of FDA (1-layer). This further demonstrates the strong potential of the theoretical FDA paradigm.
>
> Overall, the trade-off of the layer-wise strategy is effective and efficient. Moreover, the following discussion encourages us to design more practical FDA deployment strategies for multi-layer settings in future work. We thank the reviewer for the insightful comment.

---

> ### Author Response · Authors · 2025-11-21
> **Response to Reviewer snaV (Part 2)**
>
> ### **Q2: Sensitivity and Convergence Stability under Random Initialization**
>
> Thank you for this detailed and thoughtful comment regarding the sensitivity to randomness. Following your suggestions, we set five different random seeds to construct 5 groups of FDAs for experiments in our main paper. We also visualize the optimization loss curves of anchors during the optimization stage in Layer 1, 6, and 12. The link is:
> [Multi-run Angle Loss Curves](https://anonymous.4open.science/r/iclrsubmission884/multi_run_angle_loss.pdf)
>
> We report the full results in the following tables, including per-seed performance, average improvement, and standard deviation of improvements.
>
> **Multi-run Performance of FDAs on ViT-B/16**
> | Seed | FDAs (Pretrained) | FDAs (TA) | FDAs (TSV) | FDAs (WUDI) |
> |------|------------|----|-----|------|
> | 0   | 87.07 | 87.77 | 88.42 | 89.23 |
> | 42  | 87.04 | 87.68 | 88.38 | 89.20 |
> | 84  | 87.13 | 87.81 | 88.42 | 89.20 |
> | 168 | 87.10 | 87.80 | 88.38 | 89.18 |
> | 336 | 87.09 | 87.74 | 88.47 | 89.24 |
> |Avg. Improvement | **32.09** | **13.82** | **1.03** | **0.36**|
> |Std. Improvement | **0.034** | **0.052** | **0.037** | **0.024**|
>
> **Multi-run Performance of FDAs on RoBERTa-Large**
> | Seed | FDAs (Pretrained) | FDAs (TA) | FDAs (TSV) | FDAs (WUDI) |
> |------|-------------------|-----------|------------|-------------|
> | 0   | 0.6632 | 0.7220 | 0.7127 | 0.6984 |
> | 42  | 0.6631 | 0.7032 | 0.7001 | 0.6935 |
> | 84  | 0.6766 | 0.7156 | 0.7147 | 0.6980 |
> | 168 | 0.6890 | 0.7213 | 0.7143 | 0.6972 |
> | 336 | 0.6973 | 0.7260 | 0.7189 | 0.7003 |
> | Avg. Improvement | **0.302** | **0.126** | **0.052** | **0.053** |
> | Std. Improvement | **0.015** | **0.009** | **0.007** | **0.0025** |
>
> **Multi-run Performance of FDAs on LLaMA-2**
> | Seed | FDAs |
> |------|------|
> | 0   | 0.2310 |
> | 42  | 0.2310 |
> | 84  | 0.2307 |
> | 168 | 0.2255 |
> | 336 | 0.2294 |
> | Avg. Improvement | **0.0205** |
> | Std. Improvement | **0.0023** |
>
> These results demonstrate that FDA optimization exhibits highly stable convergence behavior across different random seeds. The loss curves consistently follow similar trajectories, and the final FDA solutions converge to nearly identical performance levels with only small variations. This indicates that the optimization is not sensitive to initialization and reliably reaches a stable solution. The small performance variance further confirms the robustness of the FDA training procedure.
>
> We will include the corresponding tables and convergence curves in the revised manuscript.
>
> ---
>
> ### **Q3: Layer-wise VS. Global FDA Optimization Strategy**
>
> Great suggestion! We thank the reviewer for this detailed and insightful comment. We apologize for any ambiguity in the manuscript regarding this aspect. In order to avoid introducing additional hyperparameter search, all experiments reported in the manuscript were conducted using a uniform optimization setup (same learning rate, steps, and loss weights) across all layers. Even without layer-specific tuning, FDA consistently achieves significant performance gains, demonstrating the effectiveness of the approach itself.
>
> We fully agree with the reviewer that shallow and deep layers encode different semantic information. This is a valuable observation and inspires us to explore layer-wise optimization strategies. Following this suggestion, we conducted a preliminary experiment where we reduced the number of optimization steps for shallow layers and increased them for deep layers to observe the effect on performance. We define the first six layers as the shallow part and the last six layers as the deep part. The results on ViT-B/32 are reported below:
>
> | Strategy                 | shallow-400-deep-1200 | shallow-800-deep-1200 | shallow-1200-deep-400 | shallow-1200-deep-800 | shallow-800-deep-1600 | baseline |
> |---------------------------|----------------------|----------------------|----------------------|----------------------|----------------------|----------|
> | Avg Performance          | 82.57                | 82.84                | 79.90                | 81.90                | 83.43                | 83.03    |
>
> As shown in the table, the impact of reducing optimization steps in shallow layers on performance is much smaller than that in deep layers. Given a fixed total number of iterations, increasing the number of optimization steps of FDAs in deep layers helps improve performance. Therefore, designing an appropriate layer-wise optimization strategy is also a promising direction.

---

> ### Author Response · Authors · 2025-11-21
> **Response to Reviewer snaV (Part 3)**
>
> ### **Q4: Practicality of FDA for Large-Scale Transformers**
>
> Thanks for this comment. We agree that reporting the computation time and memory cost of FDA on large-scale Transformers is crucial to demonstrate its practicality. Following the reviewer’s suggestion, we measure the computation time (in minutes) and GPU memory usage (in MB) for FDA and Prodistill on LLaMA-2-13B (RTX 5880), as shown in the table below:
>
> | Method              | Time (min) | Memory (MB) |
> |--------------------|------------|-------------|
> | Prodistill          | 532        | 33522       |
> | FDA construction (per-layer)    | 378        | 17678       |
> | FDA optimization (per-layer)   | 37         | 13192       |
>
> As the table shows, FDA construction and optimization require substantially less time and memory than Prodistill, indicating that FDA is practical even on large-scale models.
>
> Furthermore, thanks to the layer-wise optimization strategy, the FDA construction for all layers can be performed in parallel. For example, on an 8-GPU setup with RTX 4090 cards, performing 5-way parallelization reduces the total FDA construction time to approximately 48 minutes, while the FDA optimization process can be completed in only about 5 minutes.
>
> This demonstrates that FDA is both practical and highly scalable for large Transformers.
>
> ---
>
> ### **Q5: Improvement Difference between CV and NLP Tasks**
>
> We thank the reviewer for carefully noticing the difference in absolute gains between CV and NLP tasks. While the raw numbers seem to suggest larger improvements on NLP models, this is mainly due to scale differences in the original performance.
>
> To more fairly compare across modalities, we computed the percentage improvement over the task arithmetic (TA) baseline for each model. The results are shown below:
>
> | Model         | Individual | TA     | FDA    | TA-%   | FDA-%  | Improvement-% |
> |---------------|-----------|--------|--------|--------|--------|----------------|
> | ViT-B/32      | 90.52     | 69.16  | 83.75  | 0.7640 | 0.9252 | 0.1612         |
> | ViT-B/16      | 92.65     | 73.94  | 87.26  | 0.7981 | 0.9418 | 0.1438         |
> | ViT-L/14      | 94.13     | 84.54  | 91.05  | 0.8981 | 0.9673 | 0.0692         |
> | RoBERTa-base  | 0.8531    | 0.6267 | 0.6773 | 0.7346 | 0.7939 | 0.0593         |
> | RoBERTa-large | 0.8754    | 0.5918 | 0.6827 | 0.6760 | 0.7799 | 0.1038         |
>
> Thus, the average percentage improvements are 0.125 (Vision) and 0.082 (NLP), respectively.
>
> As shown, the relative gains of FDA are actually comparable across CV and NLP tasks. The apparent discrepancy in absolute numbers is primarily due to the scale difference in reporting metrics (e.g., accuracy vs. GLUE-style scores). When measured proportionally, FDA provides consistent improvements across both modalities rather than favoring NLP.
>
> We will include these results in our manuscript.
>
> ---
>
> ### **Q6: Suggestions for the Story**
>
> We are deeply appreciative of the reviewer's suggestion. We apologize for any confusion caused by our presentation. We promise to keep improving the clarity of our manuscript. In the response, we would also like to further clarify the key motivation and goal of FDA.
>
>
> The core motivation of FDA is to bridge the gap between multi-task learning, where knowledge integration naturally takes place in the input space, and model merging, where integration is typically restricted to the parameter space without access to real data. This motivates us to construct a set of synthetic points in the input space that do not rely on real data, yet still capture task-specific knowledge. Ideally, jointly training on these points should reproduce the effect of multi-task learning. However, constructing such points directly is non-trivial.
>
>
> At the same time, task vectors provide a parameter-shift direction relative to the pretrained model, and prior work has shown that moving along this direction can reproduce downstream task performance. Therefore, aligning the induced gradients of synthetic inputs with this task-vector direction provides a principled and well-founded objective for constructing such inputs. We refer to the resulting synthetic inputs as functional dual anchors (FDAs).
>
> These constructed FDAs can be treated as finetuning data and used to jointly update the pretrained parameters in a manner analogous to multi-task learning. Our empirical results show that FDAs significantly improve multi-task performance compared to task arithmetic (TA). Importantly, the task vectors used by TA are exactly the optimization target used by FDA. In other words, FDAs rely solely on the downstream checkpoints and the pretrained model, without requiring access to real data. This empirical evidence further supports that the FDA paradigm is both effective and promising.
>
> Following the reviewer's suggestion, we will improve our presentation to make it more clear and well-motivated.

---

> > ### Author Response · Authors · 2025-11-27
> > **Reply to Reviewer snaV**
> >
> > Dear Reviewer snaV,
> >
> > Thanks again for detailed feedback. We have carefully addressed all points and provided additional experiments and analysis in our previous responses (Part 1–3). As the submission deadline is approaching, we would like to kindly check whether you have any **further questions or concerns that we can help clarify**.
> >
> > If there are additional issues you would like us to address, we are more than happy to provide supplementary analyses or experiments as soon as possible.
> >
> > Thanks for your time and for helping us improve the quality of our work.
> >
> > Best regards,
> > Authors.

---

### Official Review · Reviewer_aJ6z · 2025-10-30

**Soundness:** 3
**Presentation:** 3
**Contribution:** 2
**Rating:** 6
**Confidence:** 3

**Summary:**

This paper propose a framework named Functional Dual Anchors (FDAs) to optimize synthetic inputs whose induced gradients align with task vectors, capturing task-specific functional shifts relative to the pretrained model. In addition, the authors introduce a principled initialization scheme and show that FDAs are complementary to parameter-centric model merging. Comprehensive experiments show the effectiveness of FDAs.

**Strengths:**

This paper provide a new way to understand the merging process. By reinterpreting task vectors as gradients induced by synthetic inputs, FDAs bridge the gap between multi-task learning and post-hoc model merging, offering a new functional perspective for knowledge consolidation. The discussions are sound and insightful.

**Weaknesses:**

1. In lines 55-56, the authors claimed that 'we shift the merging process into the input space, where representations can naturally capture task-specific variations.' Why does the 'merging process in the input space' can 'naturally' capture task specific variantions?

2. The whole pipeline seems to be computationally intensive. What's the exact learning time? Compared to baselines, will the performance increase be enough to offset the increase in computing complexity?

3. In line 53, the authors claimed that 'FDAs capture the analogous knowledge in the input space through their induced gradients', which sounds novel, but why modeling knowledge in such a way? Is it necessary?

4. Are the learned anchors transferable? Or we need to learn such anchors model by model?

5. Will the number of models in a merging process influence the performance of your method? To what degree does it influence the performance?

6. Will this method still work well when the models in a merging process do not share a same starting point? i.e., when they are not tuned from a same model checkpoint.

7. The anchors are model level anchors (e.g., see your Algorithm 1, lines 221-227, for each model i you optimized several anchors), which means as the number of models increase, the learning time will also increase. In addition, how to make sure that there's no conflict between different anchors, and how to ensure the diversity of the optimized anchors?

**Questions:**

See weaknesses. I'm welling to increase the rating once my concerns are well addressed.

---

> ### Author Response · Authors · 2025-11-21
> **Response to Reviewer aJ6z (Part 1)**
>
> ### **Weakness 1 & 3: Why does input-representation space merging "naturally" capture task-specific variations? Why model knowledge via FDAs?**
>
> Thanks for these two insightful questions, and we apologize for the ambiguity in the original presentation. Below, we clarify both questions through our motivation, underlying principles, and empirical evidence.
>
> **First, why can input-space merging naturally capture task-specific variations?**
> We provide two lines of evidence:
>
> 1. **From the perspective of multi-task learning vs. model merging:**
>    Multi-task learning integrates task-specific variations in the input-representation space: tasks are optimized jointly by combining their task-specific data, and the model resolves task interactions through gradients induced by the inputs. By contrast, prior model-merging methods operate purely in the parameter space, capturing task variations only via parameter offsets (task vectors). Empirically, multi-task learning consistently outperforms parameter-space merging, suggesting that the input-representation space is a more natural and expressive domain for encoding task-specific differences.
>
> 2. **From the perspective of modeling difficulty:**
>    Modeling the input space is generally easier than modeling the parameter space, as the input space is more structured and semantically aligned. This perspective has been well-established and empirically validated in dataset distillation [1,2], iterative teaching [3,4], dataset condensation [5,6], and continual learning [7,8]. These studies further support the use of input-space representations as an effective medium for expressing task-specific knowledge.
>
>
> **Second, why do we model task knowledge via induced-gradient alignment, and is it necessary?**
>
> The key motivation of FDA is to bridge the gap between multi-task learning and model merging, where knowledge integration inherently happens in the input space, and model merging is typically confined to the parameter space without access to the real data. This motivates us to construct a group of points in the input space without access to the real data. These points should capture the task-specific knowledge; that is, jointly training on these points can acquire multi-task abilities.
>
> The construction of such points is non-trivial. Note that task vectors provide a parameter-shifting direction relative to the pretrained model, and moving along this direction can reproduce the downstream model’s performance as shown in previous work [9]. Thus, aligning the induced gradients with this direction provides a principled and reasonable objective to guide the construction of input points. We call such constructed synthetic input points **functional dual anchors (FDAs)**.
>
> The constructed points (FDAs) can be treated as finetuning data and used to jointly optimize the pretrained parameters similar to multi-task training. Our empirical results show that FDAs significantly improve multi-task performance compared to the task arithmetic (TA) method. It is worth noting that the task vectors used in the TA method are exactly the construction objective of FDA. In other words, FDAs use only the parameter information from the downstream models and the pretrained model. This empirical evidence further supports that the FDA paradigm is effective and promising.
>
> Overall, the empirical success of FDA further suggests that the input space provides a natural medium for capturing task-specific variations. Our current FDA formulation offers one effective way to project and exploit task knowledge in the input space. However, we emphasize that FDA is not the only possible approach for achieving this goal. We hope that our work can inspire future research to explore more efficient or theoretically grounded mechanisms for modeling task knowledge in the input space.
>
> **References:**
> 1. Wang T, Zhu J Y, Torralba A, et al. *Dataset distillation.* arXiv preprint arXiv:1811.10959, 2018.
> 2. Cazenavette G, Wang T, Torralba A, et al. *Dataset distillation by matching training trajectories.* CVPR 2022: 4750–4759.
> 3. Liu W, Dai B, Humayun A, et al. *Iterative machine teaching.* ICML 2017: 2149–2158.
> 4. Qiu Z, Liu W, Xiao T Z, et al. *Iterative teaching by data hallucination.* arXiv preprint arXiv:2210.17467, 2022.
> 5. Zhao B, Mopuri K R, Bilen H. *Dataset condensation with gradient matching.* arXiv preprint arXiv:2006.05929, 2020.
> 6. Bo Zhao and Hakan Bilen. *Dataset condensation with distribution matching.* WACV 2023.
> 7. Shin H, Lee J K, Kim J, et al. *Continual learning with deep generative replay.* NeurIPS 2017, 30.
> 8. Yu L, Hu T, Hong L, et al. *Continual learning by modeling intra-class variation.* arXiv preprint arXiv:2210.05398, 2022.
> 9. Wortsman M, Ilharco G, Kim J W, et al. *Robust fine-tuning of zero-shot models.* CVPR 2022: 7959–7971.

---

> ### Author Response · Authors · 2025-11-21
> **Response to Reviewer aJ6z (Part 2)**
>
> ### **Weakness 2: Exact Learning Time and Worthiness of FDA's Computing Complexity**
>
> Great question! We agree with the reviewer that a detailed study of computational complexity is important. However, before delving into details, we want to also emphasize that FDA merging only consumes a one-time training cost, and the resulting merged model does not have any inference overhead. Therefore, as long as the computational overhead is not significantly larger and the performance gain is nontrivial, it will still be desirable in practice.
>
> First, reporting the exact wall-clock time and memory usage is indeed essential for assessing the practical viability of FDA. In response, we provide a detailed comparison of both the wall-clock time and GPU memory cost between FDA and the baselines, and further analyze their computational behaviors. Note that the computational cost of TA is negligible and thus omitted.
>
> | Method                        | ViT-B/16 Time (min) | ViT-B/16 Memory (MB) | RoBERTa-large Time (min) | RoBERTa-large Memory (MB) |
> |-------------------------------|-------------------|-------------------|------------------------|-------------------------|
> | TSV                           | 0.21              | 5082              | 1.3                    | 15026                   |
> | WUDI                          | 1.4               | 2948              | 5.4                    | 17500                   |
> | AdaMerging                     | 156               | 22908             | 22                     | 18984                   |
> | Prodistill                     | 11                | 2628              | 36                     | 4426                    |
> | FDA-construction (per layer)  | 22.3              | 5920              | 5                      | 986                     |
> | FDA-optimization (per layer)  | 0.92              | 3994              | 0.06                   | 2386                    |
>
> As shown in the above table, thanks to the layer-wise strategy, the memory footprint of FDA during both the construction and optimization phases remains consistently low, allowing it to run comfortably on commonly available GPUs such as the RTX 3090 or 4090.
>
> For the time usage, FDA is highly efficient in the optimization phase, while the construction phase is relatively slower. However, it remains practical for two reasons:
>
> 1. Thanks to the layer-wise strategy, FDAs for different layers can be constructed independently, enabling efficient multi-GPU parallelization.
> 2. For the same pre-trained model and tasks, the constructed FDAs can be reused once created, and the FDA optimization phase is highly efficient.
>
> For example, on an easily accessible 8×RTX 4090 server, using 8-GPU parallelization, constructing FDAs for the whole ViT-B/16 model requires only two runs, reducing the construction time to approximately 45 minutes, while the optimization phase takes only about 2 minutes.
>
>
> Second, to illustrate the worthiness of FDA's computing complexity, we extend the computation budget of the baselines and evaluate whether increased training time improves their performance to the level achieved by FDA. This directly tests whether FDA’s performance gains stem from its algorithmic design rather than simply from using more compute.
>
> Concretely, we select two representative state-of-the-art model merging methods from different categories: WUDI (data-free) and Prodistill (data-dependent), and increase their training iterations while keeping all other hyperparameters fixed. We report the results in the following table. The "baseline" denotes the default settings of these methods. We increase the number of training iterations by 5× and 10×, thereby proportionally expanding the computation time (and computational complexity) of the baselines.
>
> | Method       | baseline | 5× computation time | 10× computation time |
> |-------------|---------|-------------------|--------------------|
> | Prodistill  | 88.92   | 89.15             | 89.12              |
> | WUDI        | 88.85   | 88.86             | 88.67              |
> | FDAs (WUDI)  | 89.23   | -                 | -                  |
>
> As shown in the table, simply increasing the computation time of the baselines does not lead to meaningful performance improvement.
>
> - For Prodistill, expanding the computation budget by 5× or even 10× results in virtually no gain (88.92 → 89.15 → 89.12), indicating that its performance saturates quickly and additional computation does not translate into better merging quality.
> - Similarly, WUDI shows almost no improvement when its computation is increased, and even exhibits slight degradation at 10× computation time (88.85 → 88.86 → 88.67).
>
> In contrast, applying FDA to WUDI yields a clear and stable improvement (89.23). This result demonstrates that the performance gains provided by FDA do not arise from merely allocating more computation, but instead stem from the algorithmic advantage introduced by FDA’s input-representation formulation.

---

> ### Author Response · Authors · 2025-11-21
> **Response to Reviewer aJ6z (Part 3)**
>
> ### **Weakness 4 & 6: Are FDAs Transferable?**
>
> Thanks for these two insightful and closely related questions. The transferability of FDA essentially concerns whether the knowledge encoded by the anchors can generalize across models that share the same architecture but differ in their pretrained parameters.
>
> For models that originate from the same pretrained checkpoint, FDA anchors are constructed only once for each task. Moreover, when the model is initialized from multi-task parameters produced by other model-merging methods, FDA remains effective, as demonstrated in our experiments. These results indicate that FDA exhibits good transferability under shared initialization.
>
> However, for models starting from different pretrained checkpoints, this setting has not been explored in existing model-merging literature, and current methods, including ours, are not designed for this more challenging scenario. However, in order to better address the reviewer's concerns, we still conducted a preliminary study to examine the transferability of FDAs under this setting.
>
> One of the key challenges is that, for general large-scale architectures, obtaining multiple diverse yet high-quality pretrained checkpoints requires prohibitively expensive pretraining. To circumvent this issue, we designed a controlled toy experiment: we constructed an MLP architecture, pretrained it on CIFAR-10, and then randomly partitioned CIFAR-10 into five binary classification subsets as downstream tasks for finetuning. Within this controlled setting, we analyzed the transferability of FDAs.
>
> **Results for seed-0**
>
> | Method          | Task 0 | Task 1 | Task 2 | Task 3 | Task 4 | Average |
> |-----------------|--------|--------|--------|--------|--------|---------|
> | Individual      | 66.00  | 84.55  | 77.90  | 91.05  | 73.50  | 78.60   |
> | FDAs-seed-0     | 65.80  | 83.50  | 77.30  | 84.65  | 72.55  | 76.76   |
> | FDAs-seed-42    | 65.55  | 83.25  | 77.25  | 83.65  | 72.30  | 76.40   |
>
> **Results for seed-42**
>
> | Method          | Task 0 | Task 1 | Task 2 | Task 3 | Task 4 | Average |
> |-----------------|--------|--------|--------|--------|--------|---------|
> | Individual      | 65.90  | 84.55  | 79.50  | 91.05  | 72.95  | 78.79   |
> | FDAs-seed-42    | 65.75  | 83.10  | 78.40  | 88.65  | 72.25  | 77.63   |
> | FDAs-seed-0     | 65.50  | 82.45  | 78.15  | 88.50  | 71.40  | 77.20   |
>
>
> Across the toy experiment, we observe that FDAs constructed from one pretrained checkpoint can still improve multi-task performance when transferred to a model initialized from a different pretrained checkpoint. This demonstrates that FDA possesses initial transferability potential even under heterogeneous initialization.
>
> However, we also note that transferred FDAs consistently perform slightly worse than FDAs reconstructed directly from the target checkpoint. This indicates that while FDA captures some generalizable task structure, part of the information encoded in FDAs remains tied to the specific pretrained initialization.
>
> Overall, these results suggest that FDA is promising in this challenging setting, but further improving FDA's transferability remains an open direction for future work.
>
> ---
> ### **Weakness 5: Effect of the Number of Tasks on Merging Performance**
>
> Thanks for the insightful question. In general, the performance of both model merging and multi-task learning tends to degrade as the number of tasks increases. This is because model capacity is limited, and introducing more tasks brings more potential conflicts. Following your suggestion, we examined how the performance changes when merging different numbers of tasks. For comparison, we also consider the performance changes of task arithmetic (TA) method. Since the set of merged tasks affects the average performance of the finetuned models themselves, directly comparing raw accuracy values can be misleading. Therefore, we report the curve by ratio = (Average Multi-task Performance) / (Average Performance of the Finetuned Models), which provides a normalized and fair measure of how well each merging method preserves the original task performance across different task counts. The curve is visualized in the link: https://anonymous.4open.science/r/iclrsubmission884/Task_Scaling_Effect.png. From the visualization, both FDA and TA are affected by the increase in the number of tasks. However, FDA exhibits significantly smaller fluctuations in performance, indicating that it is more robust to task scaling compared to TA.

---

> ### Author Response · Authors · 2025-11-21
> **Response to Reviewer aJ6z (Part 4)**
>
> ### **Weakness 7：The Conflict and Diversity of different FDAs**
> Thanks for raisng these constructive questions.
>
> As you correctly pointed out, FDAs are model-level (or task-level) anchors. Therefore, when the number of tasks increases, additional FDAs need to be constructed, which indeed introduces extra overhead. However, once constructed, FDAs can be reused indefinitely; the subsequent parameter updates based on FDA guidance are extremely efficient.
>
> Regarding potential conflicts, as the number of tasks grows, inter-task conflicts are generally unavoidable. This is a well-known challenge in both multi-task learning and model merging. Nevertheless, our experiments show that conflicts among FDAs are significantly milder than conflicts directly operating in parameter space, demonstrating the advantage of projecting task knowledge into the input-space paradigm. Moreover, we compute the average pair-wise cosine similarity between the current task-level FDAs to observe the current conflicts. The lower the similarity, the closer the vectors are to being orthogonal, which suggests that potential conflicts may be smaller. We report the results in the follow table:
>
> FDAs from ViT-B/32 | layer 1 | layer 6 | layer 12
> --- | --- | --- | ---
> avg cos sim | 0.0583 | 0.0255 | 0.0500
>
> Thus, the task-level FDAs exhibit consistently low pairwise cosine similarity across layers, indicating that they are well separated in the input space. This suggests that under the current paradigm, FDAs constructed for different tasks naturally exhibit low levels of conflict.
>
> For diversity, we consider two perspectives.
> (1) **Inter-task diversity**: FDAs are defined by task vectors. Therefore, as long as task vectors themselves are diverse, well-optimized FDAs naturally inherit this diversity. A t-SNE visualization (https://anonymous.4open.science/r/iclrsubmission884/Visualization_3_of_anchors.jpeg) of FDAs optimized on ViT-B/32 clearly shows that FDAs from different tasks form distinct clusters.
>
> (2) **Intra-task diversity**: One intuitive way to increase the diversity of intra-task anchors is to reduce the similarity between this anchors. Motivated by your suggestion, during FDA construction we apply such a uniformity regularizer, encouraging anchors to spread evenly on the hypersphere. The formulation is as follows:
> $L_{\text{uni}}(X) = \frac{1}{N(N-1)} \sum_{i \ne j} \left( \hat{x}_i^\top \hat{x}_j \right)^2, \hat{x}_i =\frac{x_i}{\|x_i\|}$, where $X\in  R^{N\times d}$; $d$ is the length of the anchor.
> To explore the effect of the diversity, we consider three different regularization coefficients: $0.02, 0.1, 0.5$. We report the results of ViT-B/32 in the following Table:
>
> regularization coefficient | 0 | 0.02 | 0.1 | 0.5
> --- | --- | --- | --- | ---
> Avg Performance | 83.03 | 83.18 | 83.17 | 83.15
>
> As shown in the table, adding this uniformity regularizer produces a slight improvement over the baseline setting without regularization. This suggests that encouraging intra-task diversity indeed helps construct more effective anchors. Although the gain is modest in this initial study, it provides evidence that diversity is a meaningful factor in FDA construction.
>
> We believe this direction is promising, and we thank the reviewer for raising this insightful suggestion. It has inspired us to further investigate more principled diversity-promoting mechanisms for constructing anchors in future work.

---

> ### Comment · Reviewer_aJ6z · 2025-11-25
>
> Thanks for the authors' detailed response. Taking both the content of the paper and the authors’ response, I believe the main novelty of this work lies in exploring a new possibility of input-level model merging. Although this pipeline may face some potential issues in practical deployment (e.g., efficiency and scalability), I still consider it valuable, and I would say this paper meet the acceptance bar of ICLR. Therefore, I would like to keep my positive score unchanged. Good luck with your submission.

---

> > ### Author Response · Authors · 2025-11-27
> > **Reply to Reviewer aJ6z**
> >
> > We sincerely thank Reviewer aJ6z for the constructive feedback throughout the review process and for the encouraging final assessment. We appreciate your recognition of the novelty of exploring input-level model merging.
> >
> > Regarding the practical considerations, we fully agree that efficiency and scalability are crucial for model merging methods in practical scenarios. In this respect, we would like to briefly clarify that FDA is designed with one-time construction cost and zero inference overhead. Moreover, the layer-wise formulation enables **parallelizable and memory-efficient training**, making the pipeline applicable with commonly available hardware.
> >
> > We will strengthen the discussion of these practical aspects in the final version of the paper to better convey the applicability and limitations of FDA in practice.
> >
> > Thank you again for your thoughtful evaluation and supportive comments.

---

### Official Review · Reviewer_3H3a · 2025-10-31

**Soundness:** 2
**Presentation:** 2
**Contribution:** 2
**Rating:** 4
**Confidence:** 3

**Summary:**

This paper presents an input-representation space model-merging framework that projects task-specific knowledge from fine-tuned checkpoints into synthetic inputs. These inputs are optimized so that their induced gradients on the pretrained model align with task vectors (parameter deltas). Experiments on vision (CLIP ViTs), language (RoBERTa), and autoregressive (LLaMA-2) models show consistent improvements over state-of-the-art baselines, with ablations validating key design choices such as initialization schemes and distance functions. However, the paper lacks a clear motivation and a solid theoretical foundation for the proposed method.

**Strengths:**

1)New approach to merging: Proposes a method that projects task knowledge into the input–representation space rather than directly manipulating parameter vectors. This connects joint multi-task training (input-centric) with post-hoc weight averaging (parameter-centric), providing an alternative design perspective.
2)Theoretically grounded initialisation: Derives closed-form dynamics for a linear encoder and shows that tail eigen-energy of the task vector slows convergence. Two simple initialisation strategies (weight-row sampling and scaled Gaussian) fall out of the theory and yield the fastest loss decrease.
3)Comprehensive empirical validation: Evaluated across multiple vision datasets, NLP benchmarks, and large language models, covering both encoder and decoder architectures. FDA-enhanced TA consistently improves performance across these task.

**Weaknesses:**

1)The motivation is unclear, and the paper lacks a clear explanation of why input representations can be used to replace task vectors in model merging.
2)Limited theoretical justification beyond linear case: All convergence claims rest on a single-layer linear encoder (Sec. 2.2); no analysis for non-convex deep nets or layer interactions. No guarantee that gradient-aligned synthetic points transfer to real-data loss basins.
3)Missing statistical significance: Reported numbers are single-run means without variance estimates; it is impossible to judge whether +0.3% gains are systematic or noise (all tables).
4)Optimisation steps vs. quality: Fig. 10 stops at 1200 steps without showing whether performance saturates or collapses later.
5)Anchor shape: token-num ablation on RoBERTa shows drop beyond 5 tokens but explanation is deferred to “closer to real shape” hypothesis without measuring actual data token statistics (Sec. 5.2; Table 8).
6)Distance-metric sensitivity under-explored: Only cosine, L1, L2 are tested; manuscript does not explain why cosine is optimal or whether the choice interacts with architecture (Sec. 5.3; Fig. 9).

**Questions:**

1)Explain why inputs can replace task vectors?
2)Explain the meaning of equation (1). What does \phi(\theta,x) and Dist(.) in equation (1) mean?

---

> ### Author Response · Authors · 2025-11-21
> **Response to Reviewer 3H3a (Part 1)**
>
> ### **Q1: Why can input representations replace task vectors? (Motivation)**
>
> Great question! We apologize for the confusion caused by our initial presentation, and we thank you for highlighting this point. In response to your comment, we have re-drawn the illustration teaser figure in our paper and updated the manuscript accordingly.
>
> Here we provide a more explicit explanation of why input representations can be used in place of task vectors for model merging, and how this connects to the overarching goal of our method.
>
> The core motivation of FDA is to **bridge the gap between multi-task learning and model merging**.
>
> **Multi-task learning naturally integrates task-specific knowledge in the input–representation space**: different tasks are jointly optimized by combining their task-specific data, and the model implicitly reconciles task conflicts through gradients induced by their inputs. This demonstrates that input representations are able to intrinsically encode the information required for knowledge integration.
>
> In contrast, **model merging operates purely in the parameter space, typically by arithmetic on task vectors**. The empirical success of these approaches indicates that the directions of task vectors contain meaningful task-specific knowledge. This knowledge is expressed as a parameter offset rather than through data.
>
> Motivated by these two observations, FDA seeks to connect these two paradigms. Instead of directly manipulating task vectors, we construct **synthetic inputs, i.e., functional dual anchors (FDAs), whose induced gradients align with the task vector**. Conceptually, this is analogous to projecting the task-specific knowledge encoded in parameters back into the input–representation space. Since FDAs live in the input space, they can be treated as finetuning data. By jointly optimizing the model using FDAs from different tasks, we effectively mimic multi-task learning in a data-free manner.
>
> The second motivation is that **modeling the input space is generally easier than modeling the parameter space, as the input space tends to be more structured**. The effectiveness of modeling the input sapce for knowledge transfer has been widely explored and empricially validated in the context of dataset distillation [1,2], iterative teaching [3,4], dataset condensation [5,6] and continual learning [7,8]. These findings collectively motivate and justify our choice to use input representations in place of task vectors.
>
> Moreover, compared to the task arithmetic (TA) method, FDA transfers the knowledge from different task-specific models back into the pretrained model more effectively, thereby yielding better multi-task performance (e.g., TA: 73.94 → FDA: 87.07 on ViT-B/16; TA: 0.5918 → FDA: 0.6632 on RoBERTa-large), as shown in Table 1 and 2 of our manuscript. This empirical success further underscores the necessity and potential of leveraging input representations for effective model merging.
>
> **Reference:**
> 1. Wang T, Zhu J Y, Torralba A, et al. Dataset distillation. arXiv:1811.10959, 2018.
> 2. Cazenavette G, Wang T, Torralba A, et al. Dataset distillation by matching training trajectories. CVPR 2022.
> 3. Liu W, Dai B, Humayun A, et al. Iterative machine teaching. ICML 2017.
> 4. Qiu Z, Liu W, Xiao T Z, et al. Iterative teaching by data hallucination. arXiv:2210.17467, 2022.
> 5. Zhao B, Mopuri K R, Bilen H. Dataset condensation with gradient matching. arXiv:2006.05929, 2020.
> 6. Zhao B, Bilen H. Dataset condensation with distribution matching. WACV 2023.
> 7. Shin H, Lee J K, Kim J, et al. Continual learning with deep generative replay. NeurIPS 2017.
> 8. Yu L, Hu T, Hong L, et al. Continual learning by modeling intra-class variation. arXiv:2210.05398, 2022.

---

> > ### Comment · Reviewer_3H3a · 2025-11-26
> >
> > Thank you for the rebuttal. Unfortunately, my main concerns remain unresolved. The motivation and novelty of the proposed approach are still not clearly articulated—specifically, how it substantively differs from conventional methods and why it provides clear advantages. Because these key issues were not convincingly addressed, I will maintain my original score.

---

> > > ### Author Response · Authors · 2025-11-27
> > > **Reply to Reviewer 3H3a**
> > >
> > > We sincerely thank Reviewer 3H3a for the follow-up comment. We appreciate your time and thoughtful evaluation throughout the review process.
> > >
> > > We understand that your main concerns remain unresolved, especially regarding the clarity of the motivation and the novelty of the proposed framework. To help us further improve the final version of the paper, may we kindly ask whether you could point out the specific aspects of the motivation or the novelty that you feel are insufficiently addressed?
> > >
> > > In our rebuttal **(Response Part 1)**, we have attempted to clarify:
> > >
> > > ● **Motivation**:
> > >
> > > We explained that FDA aims to conceptually **bridge multi-task learning (which integrates task-specific knowledge in the input-representation space) and model merging (which integrates tasks through arithmetic on parameter space).** By projecting task vectors back into the input–representation space, FDA can mimic multi-task learning through synthetic inputs rather than direct weight manipulation.
> > >
> > > Another motivation is that **modeling the input space is generally easier than modeling the parameter space.** Extensive prior works in dataset distillation, iterative teaching, and gradient matching have shown that input space is more structured.
> > >
> > > ● **Novelty:**
> > >
> > > We highlighted that FDA differs from existing merging methods by shifting the merging operation from parameter space to the input–representation space. Instead of directly manipulating task vectors (as in TA, TSVM, AWD, DOGE, WUDI), FDA constructs **synthetic inputs, i.e., functional dual anchors (FDAs), whose induced gradients align with the task vector. Then, FDAs are treated as finetuning data. By jointly optmizing the model using FDAs from different tasks, the knowledge integration process of FDAs mimic multi-task learning. Therefore, FDAs enjoys the advantage from the paradigm of Multi-task learning and provides a novel method to utilize the information encoded in the model parameters.**
> > >
> > > Moreover, **our wide empirical results suggest that FDA transfers the knowledge from different task-specific models back into the pretrained model more effectively**, thereby yielding better multi-task performance (e.g., TA: 73.94 → FDA: 87.07 on ViT-B/16; TA: 0.5918 → FDA: 0.6632 on RoBERTa-large), as shown in Table 1 and 2 of our manuscript.
> > >
> > >
> > > If these points do not sufficiently address your concerns, we would be very grateful if you could indicate which parts remain unclear.
> > >
> > > We appreciate your feedback and look forward to any further suggestions you may have.

---

> ### Author Response · Authors · 2025-11-21
> **Response to Reviewer 3H3a (Part 2)**
>
> ### **Q2: Explanation for Equation 1**
>
> We apologize for the lack of clarity in our original presentation and thanks again for pointing this out. Below, we first clarify the meanings of $\varphi(\theta, x)$ and $Dist(\cdot)$ and then explain the meaning of Equation 1.
>
> In Equation 1, $\varphi(\theta,x)$ denotes the representation produced by the model $\varphi$ with parameters $\theta$ when it processes input $x$. For example, $\varphi(\theta_i,x)$ is the representation of the downstream checkpoint (the parameter is $\theta_i$) on input $x$, while $\varphi(\theta_0, x)$ corresponds to the pretrained model. The term $Dist(\cdot)$ represents a differentiable distance function, which are used to measure the discrepancy between two representations. In our paper, all experiments adopt the cosine distance.
>
> With these definitions, Equation (1) aims to find a set of inputs $\{x_{ij}\}$ whose representation-induced gradients on the pretrained model $\theta_0$ align with the task vector $\tau_i$. The gradient is obtained by computing the representation discrepancy between $\varphi(\theta_0,x)$ and $\varphi(\theta_i,x)$ on the constructed inputs, and then backpropagating this discrepancy with respect to the pretrained parameters $\theta_0$.
>
> ---
>
> ### **Q3: The Statistical Significance**
>
> We thank the reviewer for raising this important point. Multiple runs with statistical significance are essential to justify the effectiveness of FDAs. Following your suggestions, we set five different random seeds to construct 5 groups of FDAs for experiments in our main paper. We report the full results in the following tables, including per-seed performance, average improvement, and standard deviation of improvements.
>
> **Multi-run Performance of FDAs on ViT-B/16**
> | Seed | FDAs (Pretrained) | FDAs (TA) | FDAs (TSV) | FDAs (WUDI) |
> |------|------------|----|-----|------|
> | 0   | 87.07 | 87.77 | 88.42 | 89.23 |
> | 42  | 87.04 | 87.68 | 88.38 | 89.20 |
> | 84  | 87.13 | 87.81 | 88.42 | 89.20 |
> | 168 | 87.10 | 87.80 | 88.38 | 89.18 |
> | 336 | 87.09 | 87.74 | 88.47 | 89.24 |
> |Avg. Improvement | **32.09** | **13.82** |**1.03** |**0.36**|
> |Std. Improvement | **0.034** | **0.052** | **0.037** | **0.024**|
>
> **Multi-run Performance of FDAs on RoBERTa-Large**
> | Seed | FDAs (Pretrained) | FDAs (TA) | FDAs (TSV) | FDAs (WUDI) |
> |------|-------------------|-----------|------------|-------------|
> | 0   | 0.6632 | 0.7220 | 0.7127 | 0.6984 |
> | 42  | 0.6631 | 0.7032 | 0.7001 | 0.6935 |
> | 84  | 0.6766 | 0.7156 | 0.7147 | 0.6980 |
> | 168 | 0.6890 | 0.7213 | 0.7143 | 0.6972 |
> | 336 | 0.6973 | 0.7260 | 0.7189 | 0.7003 |
> | Avg. Improvement | **0.302** | **0.126** | **0.052** | **0.053** |
> |Std. Improvement | **0.015** | **0.009** | **0.007** | **0.0025** |
>
> **Multi-run Performance of FDAs on LLaMA-2**
> | Seed | FDAs |
> |------|------|
> | 0   | 0.2310 |
> | 42  | 0.2310 |
> | 84  | 0.2307 |
> | 168 | 0.2255 |
> | 336 | 0.2294 |
> | Avg. Improvement | **0.0205** |
> | Std. Improvement | **0.0023** |
>
> These results show that the performance gains of FDA are consistent across different seeds, with small standard deviations, indicating that the improvements are systematic rather than random noise. Even for strong baselines such as WUDI, while the absolute improvement appears small, it is statistically meaningful. We will include these tables in our manuscript.

---

> ### Author Response · Authors · 2025-11-21
> **Response to Reviewer 3H3a (Part 3)**
>
> ### **Q4: Theoretical Analysis and Guarantee Beyond Linear Models**
>
> We thank the reviewer for reading our theoretical analysis and pointing out the limitations of the single-layer linear case. To focus on essential behavior, we study a single-layer linear network’s impact on our initialization scheme. This simplification ignores non-linear components, but linearizing deep neural networks to $Wx+b$ is common in model merging (e.g., DOGE [1], AWD [2], TSVM [3], WUDI [4]). Empirical success justifies this approach. Moreover, figure 4 in our manuscript shows that our linear analysis insights generalize to non-linear layers, supporting its practical effectiveness.
>
> For the single-layer setting, to further investigate the influence of multi-layer interactions, we conducted an additional study on ViT-B/32 where we constructed FDAs using two consecutive Transformer blocks instead of a single layer. This experiment allows us to examine whether the conclusions derived from the single-layer linear model continue to hold when non-linear multi-layer dependencies are introduced. We visualized the construction dynamics under different scaling coefficient $\sigma$ in the link:
> https://anonymous.4open.science/r/iclrsubmission884/Optimization_loss_for_Multlayers.png.
>
> In addition, we report below the multi-task performance of ViT-B/32 optimized using these two-layer FDAs. We simply use the optimization settings in our main paper and do not perform any hyperparameter search.
>
> | Method        | $\sigma=10.0$ | $\sigma=1.0$ | $\sigma=1e-2$ | $\sigma=1e-4$ |
> |---------------|---------------|---------------|----------------|----------------|
> | FDAs (1-layer)| 77.42         | 81.78         | 83.03          | 71.75          |
> | FDAs (2-layer)| 76.78         | 81.83         | 82.6           | 70.33          |
>
> It can be seen that the conclusions derived from the single-layer linear model consistently hold true even when extending FDA construction to a deeper, non-linear setting. These empirical results further demonstrate the effectiveness of our linear analysis.
>
> For the guarantee, we acknowledge that there is no strict theoretical guarantee that gradient-aligned synthetic points (i.e., FDAs) will perfectly transfer to the real-data loss basins. Indeed, such a guarantee remains an open challenge in deep learning. However, it is worth noting that existing parameter-based model merging methods such as task arithmetic (TA), which operate by adding or interpolating weights in the weight space, similarly lack any formal guarantee of landing in a valid or performant region of the loss landscape.
>
> In contrast, FDA performs knowledge transfer in the input-representation space, which aligns more closely with the spirit of multi-task learning: instead of manipulating parameters directly, it shapes the model’s behavior through meaningful inputs that encode task-specific information. This perspective offers a more principled and interpretable pathway for model merging.
>
> Moreover, our extensive empirical results across vision, NLP, and autoregressive models consistently show that FDA leads to significant performance gains, strongly supporting its practical validity despite the absence of a formal guarantee.
>
> **References**
>
> 1. Wei Y, Tang A, Shen L, et al. *Modeling multi-task model merging as adaptive projective gradient descent*. arXiv preprint arXiv:2501.01230, 2025.
> 2. Feng Xiong, Runxi Cheng, Wang Chen, Zhanqiu Zhang, Yiwen Guo, Chun Yuan, and Ruifeng Xu. *Multi-task model merging via adaptive weight disentanglement*. arXiv preprint arXiv:2411.18729, 2024.
> 3. Gargiulo A A, Crisostomi D, Bucarelli M S, et al. *Task singular vectors: Reducing task interference in model merging*. CVPR 2025: 18695–18705.
> 4. Cheng R, Xiong F, Wei Y, et al. *Whoever started the interference should end it: Guiding data-free model merging via task vectors*. arXiv preprint arXiv:2503.08099, 2025.
>
> ---
> ### **Q5: Optimization Steps vs. Performance**
>
> Thanks for this careful attention to our experimental section. We apologize for not providing a more clear conclusion regarding performance trends at longer optimization steps; we will include a more explicit summary of these observations in the revised manuscript. In Figure 10 of our manuscript, we have already reported FDA performance at 1600 steps. Compared to 1200 steps, the performance shows negligible difference, indicating that the model’s performance has almost saturated rather than collapsed.
>
> Additionally, we conducted experiments on ViT-B-32 and RoBERTa-base with 3000 optimization steps. We visualize the performance curves in the following link: https://anonymous.4open.science/r/iclrsubmission884/fdas_of_more_steps.png. These results further confirm that FDA performance converges with longer optimization, without any collapse.

---

> ### Author Response · Authors · 2025-11-21
> **Response to Reviewer 3H3a (Part 4)**
>
> ### **Q6: Token-num Ablation**
>
> We appreciate the reviewer’s suggestion to examine actual token-length statistics. Following this advice, we computed token lengths on the GLUE validation sets using our tokenizer. The results (avg/min/max per sample) are:
>
> | task | cola | sst2 | mrpc | stsb | qqp | mnli | qnli | rte |
> |------|------|------|------|------|------|-------|-------|------|
> | avg seq len | 11.62 | 26.01 | 54.25 | 32.28 | 31.05 | 39.85 | 51.98 | 65.83 |
> | min seq len | 5 | 5 | 26 | 12 | 11 | 8 | 16 | 19 |
> | max seq len | 36 | 63 | 84 | 91 | 128 | 128 | 128 | 128 |
>
> These statistics do not show a clear trend that explains why RoBERTa performance drops beyond $token\_num=5$. However, please note that if we conduct merging *without STS-B*, the average performance on other datasets consistently increases with larger $token\_num$. Thus, this observation does not affect the main conclusion: FDA consistently improves merging performance across tasks. The token-number sensitivity on the specific dataset seems to be a dataset-specific phenomenon.
>
> Rather than over-interpreting this behavior, we now treat $token\_num$ as a hyperparameter that can be selected via standard validation protocols such as cross-validation. Importantly, as shown in our ablation study, FDA is generally robust to the choice of $token\_num$. Accordingly, we will revise the manuscript to remove the earlier “closer-to-real-shape” hypothesis and include the above exploration. We hope this additional empirical study can clarify the reviewer's concerns.
>
> ---
>
> ### **Q7: Distance-metric Sensitivity**
>
> Thanks for this insightful comment. Following the suggestion, we expanded our experiments to include two distinct distance measures: **KL divergence** and **Hyperbolic distance**.
>
> The formulation of KL divergence is as follows:
> $$
> \mathrm{SKL}(p, q) = \frac{1}{2}\left[ \sum_{i=1}^D p_i \log\frac{p_i}{q_i} + \sum_{i=1}^D q_i \log\frac{q_i}{p_i} \right],
> $$
> where
> $$
> p = \mathrm{softmax}\left(\frac{x/\|x\|_2}{T}\right), \qquad
> q = \mathrm{softmax}\left(\frac{y/\|y\|_2}{T}\right),
> $$
> and $x, y$ denote the measured features; $T$ is the temperature coefficient. We set $T = 0.1$.
>
> The formulation of Hyperbolic distance is:
> $$
> d(u, v) = \operatorname{arcosh}\left( 1 + \frac{2\|u - v\|^2}{(1 - \|u\|^2)(1 - \|v\|^2)} \right),
> $$
> where
> $$
> u = \mathrm{proj}(x), \qquad v = \mathrm{proj}(y),
> $$
> and
> $$
> \mathrm{proj}(x)=
> \begin{cases}
> x, & \|x\| \le c - \varepsilon,
> \dfrac{x}{\|x\|}(c - \varepsilon), & \|x\| > c - \varepsilon.
> \end{cases}
> $$
> Here, we set $c = 0.95$ and $\varepsilon = 1\mathrm{e}{-5}$.
>
> Both metrics were integrated into our FDA loss in the same way as cosine distance. We test them on ViT-B/32.
>
> | Construction \\ Adaptation | KL Divergence | Hyperbolic distance | Cosine |
> |----------------------------|---------------|----------------------|--------|
> | **KL Divergence**          | 81.70 | 83.60 | 83.18 |
> | **Hyperbolic distance**    | 82.21 | 84.21 | 83.36 |
> | **Cosine**                 | 81.61 | 83.40 | 83.05 |
>
> The loss function specified in each row is employed during construction, whereas the one in each column header is used during adaptation. From the above results, FDA constructed with different distance metrics consistently improved multi-task performance. This demonstrates the robustness and practical applicability of the FDA framework itself, *regardless of the specific choice of distance*.
>
> In our main paper, we adopt cosine distance as the default metric mainly because it has been widely validated in prior representation learning works (SimCLR [1], MoCo [2], BYOL [3], and CLIP [4]) and it is very simple to deploy in practice without additional hyperparameter tuning.
>
> Interestingly, thanks to the reviewer’s suggestion, we discovered that the version of FDA constructed with **hyperbolic distance** achieves even stronger performance in our extended experiments. This finding highlights the potential of exploring more advanced geometric distances within the FDA framework. We view this as a promising direction for future research and will include both the new results and corresponding discussion in the revised manuscript.
>
> **References:**
>
> 1. Chen, Ting, et al. *A simple framework for contrastive learning of visual representations.* ICML, 2020.
> 2. He, Kaiming, et al. *Momentum contrast for unsupervised visual representation learning.* CVPR, 2020.
> 3. Grill, Jean-Bastien, et al. *Bootstrap your own latent.* NeurIPS, 2020.
> 4. Radford, Alec, et al. *Learning transferable visual models from natural language supervision.* ICML, 2021.

---

### Official Review · Reviewer_L7dz · 2025-11-01

**Soundness:** 4
**Presentation:** 3
**Contribution:** 3
**Rating:** 4
**Confidence:** 3

**Summary:**

This paper introduces Functional Dual Anchors (FDAs), a novel framework for model merging that shifts the focus from the conventional parameter space to the input-representation space. The core idea is to construct a set of synthetic inputs (FDAs) for each downstream task, such that the gradients they induce on the pretrained model align with the corresponding task vector. This approach allows FDAs to encode task-specific knowledge in the input space, serving either as a standalone merging strategy or as a complementary refinement for existing parameter-centric methods.

**Strengths:**

1. The paper provides a novel and insightful perspective on model merging.
2. The proposed method is well-motivated by a solid theoretical analysis.
3. The experimental validation is comprehensive and robust, convincingly demonstrating FDA's effectiveness.

**Weaknesses:**

1. The FDA construction process involves a nested optimization problem that requires computing second-order gradients, leading to a significant computational overhead. Although the layer-wise strategy makes it tractable, the method is inherently more expensive than one-shot approaches like TA or WUDI, and the paper lacks a quantitative analysis of this extra cost, which raises concerns about its practical utility.
2. While FDAs show the ability to enhance existing methods, the performance gains on strong state-of-the-art baselines like WUDI are sometimes marginal.
3. The framework introduces a considerable number of new hyperparameters, including the number of anchors, token numbers, the scaling coefficient for initialization, optimization steps, and the choice of distance function. This complexity can make the method more difficult to tune and apply in practice compared to simpler merging algorithms.

**Questions:**

1. Could you provide a quantitative comparison of the computational overhead? For instance, what is the total wall-clock time or the number of FLOPs required to merge a set of models using your method (including both construction and optimization stages) compared to baselines like TA, TSV, and WUDI under the same hardware setup?
2. Have you attempted to visualize the constructed FDAs for vision tasks (e.g., the input to a ViT)? It would be highly insightful to see whether they resemble noisy natural images, abstract textures, or something else entirely, as this could provide a more intuitive understanding of the knowledge they capture.
3. It is noted that the performance gains on strong baselines like WUDI can be marginal. This raises a practical question regarding the tuning effort: was this small improvement achieved using a default set of hyperparameters, or did it necessitate extensive, baseline-specific tuning? More broadly, how would you advise a practitioner to reasonably select hyperparameters like the number of anchors (n) and token_num in a real-world scenario, given that the tuning process itself is computationally expensive?

---

> ### Author Response · Authors · 2025-11-21
> **Response to Reviewer L7dz (Part 1)**
>
> ### **Q1: Practical Utility of FDAs by a Quantitative Comparison**
>
> We sincerely thank the reviewer for this constructive comment. As FDA requires constructing a group of dual anchors in the input space, rather than directly using the task vectors, the extra computation in this construction phase is unavoidable. Thus, we fully agree with the reviewer that a quantitative analysis of this cost is crucial for illustrating the practical utility of FDA.
>
> Following the reviewer's suggestion, we report the time and GPU memory usage required to construct FDAs for a single layer of ViT-B/16 and RoBERTa-large. This is because FDA adopts a layer-wise optimization strategy, and the computation for different layers can be fully parallelizable. We also list the usgae of several baseline methods. Note that the cost of TA is negligible and therefore omitted.
>
> | Method | ViT-B/16 Time (min) | ViT-B/16 Memory (MB) | RoBERTa-large Time (min) | RoBERTa-large Memory (MB) |
> |--------|----------------------|-----------------------|----------------------------|----------------------------|
> | TSV | 0.21 | 5082 | 1.3 | 15026 |
> | WUDI | 1.4 | 2948 | 5.4 | 17500 |
> | AdaMerging | 156 | 22908 | 22 | 18984 |
> | Prodistill | 11 | 2628 | 36 | 4426 |
> | FDA-construction (per layer) | 22.3 | 5920 | 5 | 986 |
> | FDA-optimization (per layer) | 0.92 | 3994 | 0.06 | 2386 |
>
> As shown in the above table, thanks to the layer-wise strategy, the memory footprint of FDA during both the construction and optimization phases remains consistently low, allowing it to run comfortably on commonly available GPUs such as the RTX 3090 or 4090. For the time usage, FDA is highly efficient in the optimization phase, while the construction phase is relatively slower. However, our method remains practical for two reasons.
>
> First, thanks to the layer-wise strategy, FDAs for different layers can be constructed independently, enabling efficient multi-GPU parallelization. Second, for the same pre-trained model and tasks, the constructed FDAs can be reused once created, and the FDA optimization phase is highly efficient. For example, on an easily accessible 8×RTX 4090 server, using 8-GPU parallelization, constructing FDAs for the whole ViT-B/16 model requires only two runs, reducing the construction time to less than 45 minutes, while the optimization phase takes only about 2 minutes.
>
> More importantly, the time usage of FDAs will not affect the inference time of the merged model. Therefore, the computational overhead of FDA is one-time training cost, and FDA's performance advantage makes it highly useful in practice.

---

> ### Author Response · Authors · 2025-11-21
> **Response to Reviewer L7dz (Part 2)**
>
> ### **Q2: Visualization of FDAs for Vision Tasks**
>
> Thanks for this insightful comment. We fully agree with the reviewer that visualizing the constrcuted FDAs for vision task is an intereseting and meaningful attempt for intuitively understanding the knowledge encoded in FDAs. Following your suggestion, we made several attempts to visualize the constructed FDAs. We consider the FDAs of ViT-B/16 on the GTSRB task. Thus, the shape of FDAs for any layer is $[64,197, 768]$, where $64$ denotes that $64$ anchors; $197$ and $768$ are the token number and the dimension of the token, respectively, which are the default settings. For the real data, the first token is denoted as the classification token. As FDAs don't contain any label information, we only consider the remaining $196$ token.
>
> **Visualization 1.**
> We first flatten each anchor (shape: $[1,196,768]$) into the shape $[1,14,14,1]$. The value in the last dimension is the L2 norm of the embedding vector (length: $768$). We also sample the internal features in the same layer of real data and visualize them by the same operation. The visualization figure can be found in the link:
> https://anonymous.4open.science/r/iclrsubmission884/Visualization_1_of_anchors.pdf
> *Note:* the higher the intensity, the closer the color is to white.
> We found that the internal features of real data become increasingly sparse as the layer depth increases. However, FDAs at different layers do not exhibit this trend.
>
> **Visualization 2.**
> As shown in Observation 2 of Section 4 in our paper, the linear subspaces spanned by the tokens of FDAs and real data are highly aligned. Inspired by this oberservation, we further visualize the embedding vectors of those tokens that exhibit high cosine similarity with anchors. For better visualization, we reshape each embedding vector (length = 768) into a matrix $32 \times 24$. The visualization can be found in the link:
> https://anonymous.4open.science/r/iclrsubmission884/Visualization_2_of_anchors.pdf. The visual patterns in the above link also do not exhibit obvious structural characteristics in either the anchor tokens or the real data features. This is consistent with our understanding of how deep learning models process information in a highly nonlinear, nonconvex manner, suggesting that the model captures knowledge at a latent level rather than through low-level visual patterns.
>
> **Visualization 3.**
> The above two attempts fail to reveal knowledge from FDAs through visualization. Instead, we examine the differences between FDAs across tasks. Specifically, we apply t-SNE to FDA from different tasks and visualize their low-dimensional embeddings. The visualization figure is also in the link:
> https://anonymous.4open.science/r/iclrsubmission884/Visualization_3_of_anchors.jpeg. The clear task-wise clustering exists in the visualization. This indicates that FDAs effectively encode task-specific information.
>
> Generally, our three visualization attempts reveal that FDAs indeed capture task-specific knowledge, but in a fundamentally different form from real data.

---

> ### Author Response · Authors · 2025-11-21
> **Response to Reviewer L7dz (Part 3)**
>
> ### **Q3: Performance Gains and Hyperparameter Settings for FDAs**
>
> Thanks for this thoughtful comment about our experiment results.
>
> First, in our experiments, for the same architecture but different initial parameters, we adopt excatly the same set of FDAs. In other words, for models initialized from pretrained parameters, TA, TSV and WUDI, we consistently apply the same set of FDAs for the adaptation process. For a given model architecture (e.g., ViT-B/32), only one set of FDAs needs to be constructed and can be reused across the above initializations. Therefore, the construction of FDAs are not tuned in a baseline-specific manner, showing the robustness of knowledge encoded in FDAs. To better address the reviewer's concerns, we will explicitly clarify this in the revised manuscript to avoid potential ambiguity.
>
> Second, we would like to clarify why the performance gains over strong baselines such as WUDI may appear small. Previous work [1, 2, 3] has shown that the performance of a model learned by multi-task learning serves as the upper bound of any model merging method. When a baseline already performs very close to this upper bound, the room for improvement becomes inherently limited. For example, on ViT-B/32, WUDI already achieves about 95.9% of the multi-task learning performance (Multi-task learning is the performance ceiling and count as 100%). Even in such a saturated regime, FDA is still able to further improve the merged model to 96.78% of the multi-task performance, demonstrating that FDA can extract additional gains even when the baseline is near its theoretical limit.
>
> Lastly, we provide further discussion on the hyperparameter settings for constructing FDAs, given their relatively high computational cost. We emphasize that the effectiveness of FDA is largely insensitive to hyperparameter choices. As shown in our ablation study, we sweep anchor numbers {32, 64, 128, 256}, token numbers {25, 50, 75, 100} and {1, 5, 10, 20}, loss functions {cosine, L1, L2}, and scaling coefficients from $10^1$ to $10^{-2}$. Across all these configurations, the constructed FDAs consistently improve multi-task performance compared to the TA baseline, demonstrating strong robustness of our FDA framework. Second, in all experiments presented in the paper, we apply the same FDA optimization strategy and the same hyperparameter settings across models with the same architecture but different sizes. Based on these findings, for broader applications, we recommend starting with the default hyperparameter configurations provided in the paper on smaller models and then extending them to larger or more general models with minimal adjustment.
>
> **Reference:**
> 1. Jin, Xisen, et al. "Dataless knowledge fusion by merging weights of language models." arXiv preprint arXiv:2212.09849 (2022).
> 2. Yadav, Prateek, et al. "Ties-merging: Resolving interference when merging models." Advances in Neural Information Processing Systems 36 (2023): 7093-7115.
> 3. Yang, Enneng, et al. "Adamerging: Adaptive model merging for multi-task learning." arXiv preprint arXiv:2310.02575 (2023).

---

> ### Comment · Reviewer_L7dz · 2025-11-26
>
> I thank the authors for the detailed rebuttal, particularly the quantitative analysis of computational costs and the visualization efforts. I have raised my score to acknowledge the technical novelty of the proposed approach.
>
> However, my primary concern regarding practicality remains. The main advantage of model merging typically lies in its operational simplicity and low computational overhead. The proposed FDA method, with its extra step of anchor construction, conceptually resembles Data-Free Knowledge Distillation rather than traditional lightweight merging. I recommend explicitly discussing this trade-off between performance gains and computational complexity in the final manuscript to better position the work.

---

> > ### Author Response · Authors · 2025-11-27
> > **Reply to Reviewer L7dz**
> >
> > We sincerely thank the reviewer for the thoughtful follow-up and for acknowledging the technical novelty of our work. We truly appreciate your constructive perspective throughout the review process.
> >
> > Regarding the important point you raised about the trade-off between performance gains and computational complexity: we fully agree that this discussion is essential for properly positioning FDA. Following your suggestion, we expanded our analysis to more clearly compare the computational overhead of FDA with baselines when given additional compute.
> >
> > | Method       | baseline | 5× computation time | 10× computation time |
> > |-------------|---------|-------------------|--------------------|
> > | Prodistill  | 88.92   | 89.15             | 89.12              |
> > | WUDI        | 88.85   | 88.86             | 88.67              |
> > | FDAs (WUDI)  | 89.23   | -                 | -                  |
> >
> > These additional results further highlight that the performance improvements brought by FDAs are commensurate with the extra computational budget and in many cases even surpass it, illustrating that FDA provides an effective trade-off between cost and accuracy.
> >
> > We will integrate this discussion into the final version of the manuscript to clearly articulate FDA’s position between lightweight merging and data-free knowledge distillation, ensuring that readers understand both the strengths and the computational considerations of our approach.
> >
> > Thank you again for your valuable comments and for helping us improve the clarity and positioning of our work.

---

> ### Comment · Reviewer_L7dz · 2025-11-27
>
> Thank you for the clarification and additional results. I am satisfied with the response and will keep my positive score. Good luck with the final submission.

---

### Meta-Review · Area_Chair_VkPk · 2026-01-03

**Summary:**

This paper proposes Functional Dual Anchors (FDAs), a framework for model merging that shifts the focus from the conventional parameter space to the input-representation space. Instead of combining task vectors in weight space, FDA constructs synthetic inputs whose induced gradients align with task vectors. This alignment allows the synthetic inputs to capture task-specific functional shifts relative to the original pretrained model. The methodology is designed to bridge the gap between post-hoc model merging and joint multi-task training. By treating FDAs as synthetic fine-tuning data, the model can be jointly optimized to consolidate knowledge from multiple tasks in a manner similar to multi-task learning. The empirical evaluation includes the ViT, RoBERTa, and LLaMA-2 models.

The outstanding issues contributing to the Reject recommendation:

+ **Practical applicability**: This issue was raised by multiple reviewers, who point to questions regarding optimization efficiency, a lack of empirical runtime cost in the original submission, and potential problems with scalability. These issues were partially addressed in rebuttal.
+ **Motivations and theoretical justification**: Two reviewers noted a lack of strong theoretical motivations and justification for the proposed approach, commenting that the theoretical analysis in the submission is limited to simple, linear-layer networks, and thus the question of applicability to deep (much less very deep) transformers remains unanswered. The author rebuttal did little to assuage these concerns.
+ **Comparison with the state-of-the-art**: A reviewer questioned the omission of merging results with more models, and upon closer insepection the results presented in the submission are not well aligned with the state-of-the-art literature. More precisely, all recent papers on model merging -- including TSV and TSV-M -- include results on ViT model merging for 8, 14, and 20 models. Moreover, there are more recent works based on parameter-space analysis that report results comparable to or superior to the results of FDA [1]. Finally, there are some discrepancies in the results reported in the submitted manuscript for TSV and those reported in the original paper.

Reviewers agree that the paper has some very interesting ideas in it, but the above outstanding issues outweigh these considerations and the recommendation is thus to Reject.

[1] Marczak et al. "No task left behind: Isotropic model merging with common and task-specific subspaces." ICML, 2025.

**Reviewer Concerns:**

Reviewer concerns regarding computational efficiency (Reviewers aJ6z, snaV, L7dz) were the best addressed in the author rebuttal. However, questions regarding theoretical motivations and justifications (reviewers 3H3a and aJ6z) were not adequately addressed. Most importantly, the query from Reviewer aJ6z regarding the scalability to merging more tasks was inadequate as it does not provide a thorough comparison with recent methods on merging of more tasks.

**Reviewer Scores:**

+ **R1 (L7dz)**: Mainly concerned about theoretical motivations, hyperparameter settings, efficiency, and comparison with recent methods. They seem to have been at least partially convinced by the rebuttal, but only to the point of maintaining their score.
+ **R2 (3H3a)**: Stated in the discussion period they were unconvinced by the theoretical motivations provided in rebuttal and clearly state they will keep their original score.
+ **R3 (aJ6z)**: The only reviewer sufficiently convinced by the rebuttal to improve their score to Marginally Above (as stated during the discussion). Main concerns regarded efficiency, which was the issue best addressed in rebuttal.
+ **R4 (snaV)**: Another reviewer concerned about theoretical guarantees. Doubtful they would have significantly changed their score on the basis of further discussion.

---

### Decision · Program_Chairs · 2026-01-26

Reject